# CONFIDENCE-BASED FEATURE IMPUTATION FOR GRAPHS WITH PARTIALLY KNOWN FEATURES

**Daeho Um, Jiwoong Park, Seulki Park, Jin Young Choi**
Department of Electrical and Computer Engineering, ASRI
Seoul National University
{daehoum1,ptywoong,seulki.park,jychoi}@snu.ac.kr

## ABSTRACT

This paper investigates a missing feature imputation problem for graph learning tasks. Several methods have previously addressed learning tasks on graphs with missing features. However, in cases of high rates of missing features, they were unable to avoid significant performance degradation. To overcome this limitation, we introduce a novel concept of channel-wise confidence in a node feature, which is assigned to each imputed channel feature of a node for reflecting certainty of the imputation. We then design pseudo-confidence using the channel-wise shortest path distance between a missing-feature node and its nearest known-feature node to replace unavailable true confidence in an actual learning process. Based on the pseudo-confidence, we propose a novel feature imputation scheme that performs channel-wise inter-node diffusion and node-wise inter-channel propagation. The scheme can endure even at an exceedingly high missing rate (e.g., 99.5%) and it achieves state-of-the-art accuracy for both semi-supervised node classification and link prediction on various datasets containing a high rate of missing features. Codes are available at https://github.com/daehoum1/pcfi.

## 1 INTRODUCTION

In recent years, graph neural networks (GNNs) have received considerable attention and have performed outstandingly on numerous problems across multiple fields (Zhou et al., 2020; Wu et al., 2020). While various GNNs handling attributed graphs are designed for node representation (Defferrard et al., 2016; Kipf & Welling, 2016a; Veličković et al., 2017; Xu et al., 2018) and graph representation learning (Kipf & Welling, 2016b; Sun et al., 2019; Velickovic et al., 2019), GNN models typically assume that features of all nodes are fully observed. In real-world situations, however, features in graph-structured data are often partially observed, as illustrated in the following cases. First, collecting complete data for a large graph is prohibitively expensive or even impossible. Second, measurement failure is common. Third, in social networks, most users desire to protect their personal information selectively. As data security regulation continues to tighten around the world (GDPR), access to full data is expected to become increasingly difficult. Under these circumstances, most GNNs cannot be applied directly due to incomplete features.

Several methods have been proposed to solve learning tasks with graphs containing missing features (Jiang & Zhang, 2020; Chen et al., 2020; Taguchi et al., 2021), but they suffer from significant performance degradation at high rates of missing features. A recent work by (Rossi et al., 2021) demonstrated improved performance by introducing feature propagation (FP), which iteratively propagates known features among the nodes along edges. However, even FP cannot avoid a considerable accuracy drop at an extremely high missing rate (e.g., 99.5%). We assume that it is because FP takes graph diffusion through undirected edges. Consequently, in FP, message passing between two nodes occurs with the same strength regardless of the direction. Moreover, FP only diffuses observed features channel-wisely, which means that it does not consider any relationship between channels.

Therefore, to better impute missing features in a graph, we propose to consider both inter-channel and inter-node relationships so that we can effectively exploit the sparsely known features. To this end, we design an elaborate feature imputation scheme that includes two processes. The first process is the feature recovery via channel-wise inter-node diffusion, and the second is the feature

refinement via node-wise inter-channel propagation. The first process diffuses features by assigning different importance to each recovered channel feature, in contrast to usual diffusion. To this end, we introduce a novel concept of channel-wise *confidence*, which reflects the quality of channel feature recovery. This *confidence* is also used in the second process for channel feature refinement based on highly confident feature by utilizing the inter-channel correlation.

The true *confidence* in a missing channel feature is inaccessible without every actual feature. Thus, we define pseudo-confidence for use in our scheme instead of true confidence. Using channel-wise confidence further refines the less confident channel feature by aggregating the highly confident channel features in each node or through the highly confident channel features diffused from neighboring nodes.

The key contribution of our work is summarized as follows: (1) we propose a new concept of channel-wise confidence that represents the quality of a recovered channel feature. (2) We design a method to provide pseudo-confidence that can be used in place of unavailable true confidence in a missing channel feature. (3) Based on the pseudo-confidence, we propose a novel feature imputation scheme that achieves the state-of-the-art performance for node classification and link prediction even in an extremely high rate (e.g., 99.5%) of missing features.

## 2 RELATED WORK

### 2.1 LEARNING ON GRAPHS WITH MISSING NODE FEATURES

The problem with missing data has been widely investigated in the literature (Allison, 2001; Loh & Wainwright, 2011; Little & Rubin, 2019; You et al., 2020). Recently, focusing on graph-structured data with pre-defined connectivity, there have been several attempts to learn graphs with missing node features. (Monti et al., 2017) proposed recurrent multi-graph convolutional neural networks (RMGCNN) and separable RMGCNN (sRMGCNN), a scalable version of RMGCNN. Structure-attribute transformer (SAT) (Chen et al., 2020) models the joint distribution of graph structures and node attributes through distribution techniques, then completes missing node attributes. GCN for missing features (GCNMF) (Taguchi et al., 2021) adapts graph convolutional networks (GCN) (Kipf & Welling, 2016a) to graphs that contain missing node features via representing the missing features using the Gaussian mixture model. Meanwhile, a partial graph neural network (PaGNN) (Jiang & Zhang, 2020) leverages a partial message-propagation scheme that considers only known features during propagation. However, these methods experience large performance degradation when there exists a high feature missing rate. Feature propagation (FP) (Rossi et al., 2021) reconstructs missing features by diffusing known features. However, in diffusion of FP, a missing feature is formed by aggregating features from neighboring nodes regardless of whether a feature is known or inferred. Moreover, FP does not consider any interdependency among feature channels. To utilize relationships among channels, we construct a correlation matrix of recovered features and additionally refine the features.

### 2.2 DISTANCE ENCODING

Distance encoding (DE) on graphs defines extra features using distance from a node to the node set where the prediction is made. (Zhang & Chen, 2018) extracts a local enclosing subgraph around each target node pair, and uses GNN to learn graph structure features for link prediction. (Li et al., 2020) exploits structure-related features called DE that encodes distance between a node and its neighboring node set with graph-distance measures (e.g., shortest path distance or generalized PageRank scores (Li et al., 2019)). (Zhang et al., 2021) unifies the aforementioned techniques into a *labeling trick*. Heterogeneous graph neural network (HGNN) (Ji et al., 2021) proposes a heterogeneous distance encoding in consideration of multiple types of paths in enclosing subgraphs of heterogeneous graphs. Distance encoding in existing methods improves the representation power of GNNs. We use distance encoding to distinguish missing features based on the shortest path distance from a missing feature to known features in the same channel.

### 2.3 GRAPH DIFFUSION

Diffusion on graphs spreads the feature of each node to its neighboring nodes along the edges (Coifman & Lafon, 2006; Shuman et al., 2013; Guille et al., 2013). There are two types of transition matrices commonly used for diffusion on graphs: symmetric transition matrix (Kipf & Welling,

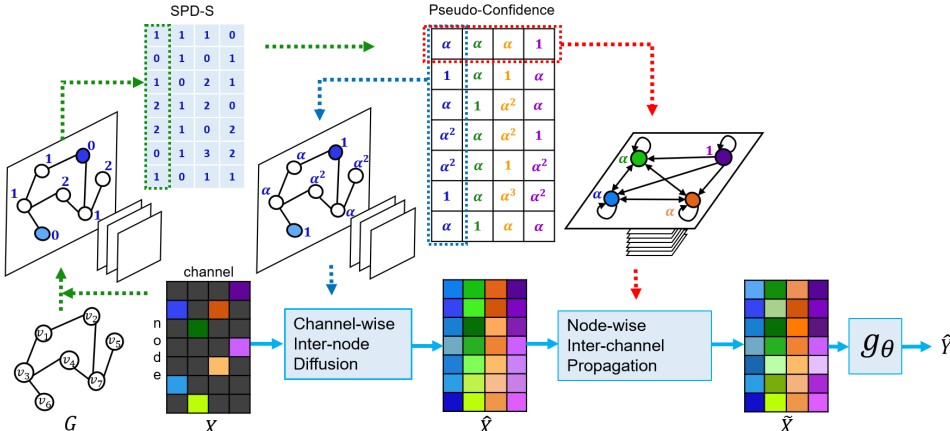

Figure 1: Overall scheme of the proposed Pseudo-Confidence-based Feature Imputation (PCFI) method. Based on the graph structure and partially known features, we calculate the channel-wise shortest path distance between a node with a missing feature and its nearest source node (SPD-S). Based on SPD-S, we determine the pseudo-confidence in the recovered feature, using a pre-determined hyper-parameter $\alpha$ $(0 < \alpha < 1)$. Pseudo-confidence plays an important role in the two stages: channel-wise Inter-node diffusion and node-wise inter-channel propagation.

2016a; Klicpera et al., 2019; Rossi et al., 2021) and random walk matrix (Page et al., 1999; Chung, 2007; Perozzi et al., 2014; Grover & Leskovec, 2016; Atwood & Towsley, 2016; Klicpera et al., 2018; Lim et al., 2021). While these matrices work well for each target task, from a node's perspective, the sum of edge weights for aggregating features is not one in general. Therefore, since features are not updated at the same scale of original features, these matrices are not suitable for missing feature recovery.

## 3 PROPOSED METHOD

### 3.1 OVERVIEW

We address a problem with **graph learning tasks containing missing node features.** To demonstrate the effectiveness of our feature imputation, we target two main graph learning tasks. The first target task, semi-supervised node classification, is to infer the labels of the unlabeled nodes from the partially known features/labels and the fully known graph structure. The second target task, link prediction, is to predict whether two nodes are likely to share a link. Figure 1 depicts the overall scheme of the proposed feature imputation. Our key idea is to assign different pseudo-confidence to each imputed channel features. To this end, the proposed imputation scheme includes two processes. The first process is the feature recovery via channel-wise inter-node diffusion, and the second is the feature refinement via node-wise inter-channel propagation. The imputed features obtained from the two processes are used for downstream tasks via off-the-shelf GNNs.

In Sec. 3.2, we begin by introducing the notations used in this paper. In Sec. 3.3, we outline the proposed PC (pseudo-confidence)-based feature imputation (PCFI) scheme that imputes missing node features. We then propose a method to determine the *pseudo-confidence* in Sec. 3.4. In Sec. 3.5, we present channel-wise inter-node diffusion that iteratively propagates known features with consideration of PC. In Sec. 3.6, we present node-wise inter-channel propagation that adjusts features based on correlation coefficients between channels.

### 3.2 NOTATIONS

**Basic notation on graphs.** An undirected connected graph is represented as $\mathcal{G} = (\mathcal{V}, \mathcal{E}, \boldsymbol{A})$ where $\mathcal{V} = \{v_i\}_{i=1}^N$ is the set of $N$ nodes, $\mathcal{E}$ is the edge set with $(v_i, v_j) \in \mathcal{E}$, and $\boldsymbol{A} \in \{0, 1\}^{N \times N}$ denotes an adjacency matrix. $\boldsymbol{X} = [x_{i,d}] \in \mathbb{R}^{N \times F}$ is a node feature matrix with N nodes and F channels, i.e., $x_{i,d}$, the $d$-th channel feature value of the node $v_i$. $\mathcal{N}(v_i)$ denotes the set of neighbors of $v_i$. Given an arbitrary matrix $\boldsymbol{M} \in \mathbb{R}^{n \times m}$, we let $\boldsymbol{M}_{i,:}$ denote the $i$-th row vector of $\boldsymbol{M}$. Similarly, we let $\boldsymbol{M}_{:,j}$ denote the $j$-th column vector of $\boldsymbol{M}$.

**Notation for graphs with missing node features.** As we assume that partial or even very few node features are known, we define $\mathcal{V}_k^{(d)}$ as a set of nodes where the $d$-th channel feature values are known ($k$ in $\mathcal{V}_k^{(d)}$ means 'known'). The set of nodes with the unknown $d$-th channel feature values is denoted by $\mathcal{V}_u^{(d)} = \mathcal{V} \setminus \mathcal{V}_k^{(d)}$. Then $\mathcal{V}_k^{(d)}$ and $\mathcal{V}_u^{(d)}$ are referred to source nodes and missing nodes, respectively. By reordering the nodes according to whether a feature value is known or not for the $d$-th channel, we can write graph signal for the $d$-th channel features and adjacency matrix as:

$$\boldsymbol{x}^{(d)} = \begin{bmatrix} \boldsymbol{x}_k^{(d)} \\ \boldsymbol{x}_u^{(d)} \end{bmatrix} \qquad \boldsymbol{A}^{(d)} = \begin{bmatrix} \boldsymbol{A}_{kk}^{(d)} & \boldsymbol{A}_{ku}^{(d)} \\ \boldsymbol{A}_{uk}^{(d)} & \boldsymbol{A}_{uu}^{(d)} \end{bmatrix}.$$

Here, $\boldsymbol{x}^{(d)}$, $\boldsymbol{x}_k^{(d)}$, and $\boldsymbol{x}_u^{(d)}$ are column vectors that represent corresponding graph signal. Since the graph is undirected, $\boldsymbol{A}^{(d)}$ is symmetric and thus $(\boldsymbol{A}_{ku}^{(d)})^\top = \boldsymbol{A}_{uk}^{(d)}$. Note that $\boldsymbol{A}^{(d)}$ is different from $\boldsymbol{A}$ due to reordering while they represent the same graph structure. $\hat{\boldsymbol{X}} = [\hat{x}_{i,d}]$ denotes recovered features for $\boldsymbol{X}$ from $\{\boldsymbol{x}_k^{(d)}\}_{d=1}^F$ and $\{\boldsymbol{A}^{(d)}\}_{d=1}^F$.

### 3.3 PC-BASED FEATURE IMPUTATION

The proposed PC-based feature imputation (PCFI) scheme leverages the shortest path distance between nodes to compute pseudo-confidence. PCFI consists of two stages: channel-wise inter-node diffusion and node-wise inter-channel propagation. The first stage, channel-wise inter-node diffusion, finds $\hat{\boldsymbol{X}}$ (recovered features for $\boldsymbol{X}$) through PC-based feature diffusion on a given graph $\mathcal{G}$. Then, the second stage, node-wise inter-channel, refines $\hat{\boldsymbol{X}}$ to the final imputed features $\tilde{\boldsymbol{X}}$ by considering PC and correlation between channels.

To perform node classification and link prediction, a GNN is trained with imputed node features $\tilde{\boldsymbol{X}}$. In this work, PCFI is designed to perform the downstream tasks well. However, since PCFI is independent of the type of learning task, PCFI is not limited to the two tasks. Therfore, it can be applied to various graph learning tasks with missing node features.

Formally, the proposed framework can be expressed as

$$\hat{\boldsymbol{X}} = f_1(\{\boldsymbol{x}_k^{(d)}\}_{d=1}^F, \{\boldsymbol{A}^{(d)}\}_{d=1}^F) \tag{1a}$$

$$\tilde{\boldsymbol{X}} = f_2(\hat{\boldsymbol{X}}) \tag{1b}$$

$$\tilde{\boldsymbol{Y}} = g_\theta(\tilde{\boldsymbol{X}}, \boldsymbol{A}), \tag{1c}$$

where $f_1$ is channel-wise inter-node diffusion, $f_2$ is node-wise inter-channel propagation, and $\hat{\boldsymbol{Y}}$ is a prediction for desired output of a given task. Here, PCFI is expressed as $f_2 \circ f_1$, and any GNN architecture can be adopted as $g_\theta$ according to the type of task.

### 3.4 PSEUDO-CONFIDENCE

We begin by defining the concept of confidence in the recovered feature $\hat{x}_{i,d}$ of a node $v_i$ for channel $d$ in the first process.

**Definition 1.** *Confidence in the recovered channel feature $\hat{x}_{i,d}$ is defined by similarity between $\hat{x}_{i,d}$ and true one $x_{i,d}$, which is a value between 0 and 1.*

Note that the feature $x_{i,d}$ of a source node is observed and thus its confidence becomes 1. When the recovered $\hat{x}_{i,d}$ is far from the true $x_{i,d}$, the confidence in $\hat{x}_{i,d}$ will decrease towards 0. However, it is a chicken and egg problem to determine $\hat{x}_{i,d}$ and its confidence. That is, the confidence in $\hat{x}_{i,d}$ is unavailable before attaining $\hat{x}_{i,d}$ according to Definition 1, whereas the proposed scheme can not yield $\hat{x}_{i,d}$ without the confidence.

To navigate this issue, instead of true confidence, we design a pseudo-confidence using the shortest path distance between a node and its nearest source node for a specific channel (SPD-S). For instance, SPD-S of the $i$-th node for the $d$-th channel feature is denoted by $\boldsymbol{S}_{i,d}$, which is calculated via

$$\boldsymbol{S}_{i,d} = s(v_i | \mathcal{V}_k^{(d)}, \boldsymbol{A}^{(d)}), \tag{2}$$

where $s(\cdot)$ yields the shortest path distance between the $i$-th node and its nearest source node in $\mathcal{V}_k^{(d)}$ on $\boldsymbol{A}^{(d)}$. It is notable that, if the $i$-th node is a source node, its nearest source node is itself, meaning $\boldsymbol{S}_{i,d}$ becomes zero. We construct SPD-S matrix $S \in \mathbb{R}^{N \times F}$ of which elements are $\boldsymbol{S}_{i,d}$.

Consider $\hat{\boldsymbol{X}} = [\hat{x}_{i,d}]$ that represents the recovered features of $\boldsymbol{X}$ with consideration of feature homophily (McPherson et al., 2001) that represents a local property on a graph (Bisgin et al., 2010; Lauw et al., 2010; Bisgin et al., 2012). Due to feature homophily, the feature similarity between any two nodes tends to increase as the shortest path distance between the two nodes decreases.

Based on feature homophily, we assume that the recovered feature $\hat{x}_{i,d}$ of a node $v_i$ more confidently becomes similar to the given feature of its nearest source node as SPD-S of $v_i$ ($\boldsymbol{S}_{i,d}$) decreases. According to the assumption, we define pseudo-confidence using SPD-S in Definition 2.

**Definition 2.** *Pseudo-confidence (PC) in $\hat{x}_{i,d}$ is defined by a function $\xi_{i,d} = \alpha^{\boldsymbol{S}_{i,d}}$ where $\alpha \in (0,1)$ is a hyper-parameter.*

By Definition 2, PC becomes 1 for $\hat{x}_{i,d} = x_{i,d}$ on source nodes. Moreover, PC decreases exponentially for a missing node features as $S_{i,d}$ increases. Likewise, PC reflects the tendency toward confidence in Definition 1. We verified that this tendency exists regardless of imputation methods via experiments on real datasets (see Figure 7 in APPENDIX). Therefore, pseudo-confidence using SPD-S is properly designed to replace confidence. To the best of our knowledge, ours is the first model that leverages a distance for graph imputation.

## 3.5 CHANNEL-WISE INTER-NODE DIFFUSION

To recover missing node features in a channel-wise manner via graph diffusion, source nodes independently propagate their features to their neighbors for each channel. Instead of simple aggregating all neighborhood features with the same weights, our scheme aggregates features with different importance according to their confidences. As a result, the recovered features of missing nodes are aggregated in low-confidence and the given features of source nodes are aggregated in high-confidence, which is our design objective. To this end, we design a novel diffusion matrix based on the pseudo-confidence.

For the design, **Definition 3** first defines 'Relative PC' that represents an amount of PC in a particular node feature relative to another node feature.

**Definition 3.** *Relative PC of $\hat{x}_{j,d}$ relative to $\hat{x}_{i,d}$ is defined by $\xi_{j/i,d} = \xi_{j,d}/\xi_{i,d} = \alpha^{\boldsymbol{S}_{j,d} - \boldsymbol{S}_{i,d}}$.*

Then, suppose that a missing node feature $x_{i,d}$ of $v_i$ aggregates features from $v_j \in \mathcal{N}(v_i)$. If $v_i$ and $v_j$ are neighborhoods to each other, the difference between SPD-S of $v_i$ and SPD-S of $v_j$ cannot exceed 1. Hence, the relative PC of a node to its neighbor can be determined using **Proposition 1**.

**Proposition 1.** *If $\boldsymbol{S}_{i,d} = m \geq 1$, $v_i$ is a missing node, then $\xi_{j/i,d}$ for $v_j \in \mathcal{N}(v_i)$ is given by*

$$\begin{aligned}
\xi_{j/i,d} &= \alpha^{-1} \ if \ \boldsymbol{S}_{i,d} > \boldsymbol{S}_{j,d}, \\
\xi_{j/i,d} &= 1 \ if \ \boldsymbol{S}_{i,d} = \boldsymbol{S}_{j,d}, \\
\xi_{j/i,d} &= \alpha \ if \ \boldsymbol{S}_{i,d} < \boldsymbol{S}_{j,d},
\end{aligned}$$

*Otherwise, $v_i$ is a source node ($\boldsymbol{S}_{i,d} = 0$), then $\xi_{j/i,d}$ for $v_j \in \mathcal{N}(v_i)$ is given by*

$$\begin{aligned}
\xi_{j/i,d} &= 1 \ if \ v_j \ is \ a \ source \ node(\boldsymbol{S}_{j,d} = 1), \\
\xi_{j/i,d} &= \alpha \ if \ v_j \ is \ a \ missing \ node(\boldsymbol{S}_{j,d} = 0).
\end{aligned}$$

The proof of **Proposition 1** is given in Appendix A.1.

Before defining a transition matrix, we temporarily reorder nodes according to whether a feature value is known for the $d$-th channel, i.e., $\boldsymbol{x}^{(d)}$ and $\boldsymbol{A}^{(d)}$ are reordered for each channel as Section 3.2 describes. After the feature diffusion stage, we order the nodes according to the original numbering.

Built on Proposition 1, we construct a weighted adjacency matrix $\boldsymbol{W}^{(d)}$ for the $d$-th channel. $\boldsymbol{W}^{(d)} \in \mathbb{R}^{N \times N}$ is defined as follows,

$$\boldsymbol{W}_{i,j}^{(d)} = \begin{cases} \xi_{j/i,d} & \text{if } i \neq j, \ \boldsymbol{A}_{i,j}^{(d)} = 1 \\ 0 & \text{if } i \neq j, \ \boldsymbol{A}_{i,j}^{(d)} = 0 \\ 1 & \text{if } i = j. \end{cases} \tag{3}$$

Note that self-loops are added to $\boldsymbol{W}^{(d)}$ with a weight of 1 so that each node can keep some of its own feature.

$\boldsymbol{W}_{i,j}^d$ is an edge weight corresponding to message passing from $v_j$ to $v_i$. Proposition 1 implies that $\alpha^{-1}$ is assigned to high-PC neighbors, 1 to same-PC, and $\alpha$ to low-PC neighbors. That is, $\boldsymbol{W}^{(d)}$ allows a node to aggregate high PC more than low PC channel features from its neighbors. Furthermore, consider message passing between two connected nodes $v_i$ and $v_j$ s.t. $\boldsymbol{W}_{i,j}^{(d)} = \xi_{j/i,d} = \alpha$. By Definition 3, $\xi_{i/j,d} = \xi_{j/i,d}^{-1}$, so that $\boldsymbol{W}_{j,i}^{(d)} = (\boldsymbol{W}_{i,j}^{(d)})^{-1} = \alpha^{-1}$. This means that message passing from a high confident node to a low confident node occurs in a large amount, while message passing in the opposite direction occurs in a small amount. The hyper-parameter $\alpha$ tunes the strength of message passing depending on the confidence.

To ensure convergence of diffusion process, we normalize $\boldsymbol{W}^{(d)}$ to $\overline{\boldsymbol{W}}^{(d)} = (\boldsymbol{D}^{(d)})^{-1}\boldsymbol{W}^{(d)}$ through row-stochastic normalization with $\boldsymbol{D}_{ii}^{(d)} = \sum_j \boldsymbol{W}_{i,j}$. Since $x_k^d$ with true feature values should be preserved, we replace the first $|\mathcal{V}^{(d)}|$ rows of $\overline{\boldsymbol{W}}^{(d)}$ with one-hot vectors indicating $\mathcal{V}_k^{(d)}$. Finally, the channel-wise inter-node diffusion matrix $\widehat{W}^{(d)}$ for the $d$-th channel is expressed as

$$\widehat{\boldsymbol{W}}^{(d)} = \begin{bmatrix} \boldsymbol{I} & \boldsymbol{0}_{ku} \\ \overline{\boldsymbol{W}}_{uk}^{(d)} & \overline{\boldsymbol{W}}_{uu}^{(d)} \end{bmatrix}, \tag{4}$$

where $\boldsymbol{I} \in \mathbb{R}^{|\mathcal{V}_k^{(d)}| \times |\mathcal{V}_k^{(d)}|}$ is an identity matrix and $\boldsymbol{0}_{ku} \in \{0\}^{|\mathcal{V}_k^{(d)}| \times |\mathcal{V}_u^{(d)}|}$ is a zero matrix. Note that $\widehat{\boldsymbol{W}}^{(d)}$ remains row-stochastic despite the replacement. An aggregation in a specific node can be regarded as a weighted sum of features on neighboring nodes. A row-stochastic matrix for transition matrix means that when a node aggregates features from its neighbors, the sum of the weights is 1. Therefore, unlike a symmetric transition matrix (Kipf & Welling, 2016a; Klicpera et al., 2019; Rossi et al., 2021) or a column-stochastic (random walk) transition matrix (Page et al., 1999; Chung, 2007; Perozzi et al., 2014; Grover & Leskovec, 2016; Atwood & Towsley, 2016; Klicpera et al., 2018; Lim et al., 2021), features of missing nodes can form at the same scale of known features. Preserving the original scale allows features to recover close to the actual features.

Now, we define channel-wise inter-node diffusion for the $d$-th channel as

$$\begin{aligned} \hat{\boldsymbol{x}}^{(d)}(0) &= \begin{bmatrix} \boldsymbol{x}_k^{(d)} \\ \boldsymbol{0}_u \end{bmatrix} \\ \hat{\boldsymbol{x}}^{(d)}(t) &= \widehat{\boldsymbol{W}}^{(d)}\hat{\boldsymbol{x}}^{(d)}(t-1), \end{aligned} \tag{5}$$

where $\hat{\boldsymbol{x}}^{(d)}(t)$ is a recovered feature vector for $\boldsymbol{x}^{(d)}$ after $t$ propagation steps, $\boldsymbol{0}_u$ is a zero-column vector of size $|\mathcal{V}_u^{(d)}|$, and $t \in [1, K]$. Here we initialize missing feature values $\boldsymbol{x}_u^{(d)}$ to zero. As $K \to \infty$, this recursion converges (the proof is provided in Appendix A.2). We approximate the steady state to $\hat{\boldsymbol{x}}^{(d)}(K)$, which is calculated by $(\widehat{\boldsymbol{W}}^{(d)})^K\hat{\boldsymbol{x}}^{(d)}(0)$ with large enough $K$. The diffusion is performed for each channel and outputs $\{\hat{\boldsymbol{x}}^{(d)}(K)\}_{d=1}^F$.

Due to the reordering of nodes for each channel before the diffusion, node indices in $\hat{\boldsymbol{x}}^{(d)}(K)$ for $d \in \{1, ..., F\}$ differ. Therefore, after unifying different ordering in each $\hat{\boldsymbol{x}}^{(d)}(K)$ according to the original order in $\boldsymbol{X}$, we concatenate all $\hat{\boldsymbol{x}}^{(d)}(K)$ along the channels into $\hat{\boldsymbol{X}}$, which is the final output in this stage.

## 3.6 NODE-WISE INTER-CHANNEL PROPAGATION

In the previous stage, we obtained $\hat{\boldsymbol{X}} = [\hat{x}_{i,d}]$ (recovered features for $\boldsymbol{X}$) via channel-wise inter-node diffusion performed separately for each channel. The proposed feature diffusion is enacted based on the graph structure and pseudo-confidence, but it does not consider dependency between channels. Since the dependency between channels can be another important factor for imputing missing node features, we develop an additional scheme to refine $\hat{\boldsymbol{X}}$ to improve the performance of downstream tasks by considering both channel correlation and pseudo-confidence. At this stage, within a node, a low-PC channel feature is refined by reflecting a high-PC channel feature according to the degree of correlation between the two channels.

We first prepare a correlation coefficient matrix $\boldsymbol{R} = [\boldsymbol{R}_{a,b}] \in \mathbb{R}^{F \times F}$, giving the correlation coefficient between each pair of channels for the proposed scheme. $\boldsymbol{R}_{a,b}$, the correlation coefficient between $\hat{\boldsymbol{X}}_{:,a}$ and $\hat{\boldsymbol{X}}_{:,b}$, is calculated by

$$\boldsymbol{R}_{a,b} = \frac{\frac{1}{N-1} \sum_{i=1}^{N} (\hat{x}_{i,a} - m_a)(\hat{x}_{i,b} - m_b)}{\sigma_a \sigma_b} \qquad (6)$$

where $m_d = \frac{1}{N} \sum_{i=1}^{N} \hat{x}_{i,d}$ and $\sigma_d = \sqrt{\frac{1}{N-1} \sum_{i=1}^{N} (\hat{x}_{i,d} - m_d)^2}$.

In this stage, unlike looking across the nodes for each channel in the previous stage, we look across the channels for each node. As the right-hand graph of Figure 1 illustrates, we define fully connected directed graphs $\{\mathcal{H}^{(i)}\}_{i=1}^{N}$ called node-wise inter-channel propagation graphs from the given graph $\mathcal{G}$. $\mathcal{H}^{(i)}$ for the $i$-th node in $\mathcal{G}$ is defined by

$$\mathcal{H}^{(i)} = (\mathcal{V}^{(i)}, \mathcal{E}^{(i)}, \boldsymbol{B}^{(i)}), \qquad (7)$$

where $\mathcal{V}^{(i)} = \{v_d^{(i)}\}_{d=1}^{F}$ is a set of nodes in $\mathcal{H}^{(i)}$, $\mathcal{E}^{(i)}$ is a set of directed edges in $\mathcal{H}^{(i)}$, and $\boldsymbol{B}^{(i)} \in \mathbb{R}^{F \times F}$ is a weighted adjacency matrix for refining $\hat{\boldsymbol{X}}_{i,:}$. To refine $\hat{x}_{i,d}$ of the $i$-th node via inter-channel propagation, we assign $\hat{x}_{i,d}$ to each $v_d^{(i)}$ as a scalar node feature for the $d$-th channel ($d \in \{1, ..., F\}$). The weights in $\mathcal{E}^{(i)}$ are given by $\boldsymbol{B}^{(i)}$ in (8).

We design $\boldsymbol{B}^{(i)}$ for inter-channel propagation in each node to achieve three goals: (1) highly correlated channels should exchange more information to each other than less correlated channels, (2) a low-PC channel feature should receive more information from other channels for refinement than a high-PC channel feature, and (3) a high PC channel feature should propagate more information to other node channels than a low PC channel feature. Based on these design goals, the weight of the directed edge from the $b$-th channel to the $a$-th channel ($\boldsymbol{B}_{a,b}^{(i)}$) in $\boldsymbol{B}^{(i)}$ is designed by

$$\boldsymbol{B}_{a,b}^{(i)} = \begin{cases} \beta(1 - \alpha^{S_{i,a}})\alpha^{S_{i,b}}\boldsymbol{R}_{a,b} & \text{if } a \neq b \\ 0 & \text{if } a = b \end{cases}, \qquad (8)$$

where $\boldsymbol{R}_{a,b}$, $\alpha^{\boldsymbol{S}_{i,b}}$, and $(1 - \alpha^{\boldsymbol{S}_{i,a}})$ are the terms fore meeting design goals (1), (2), and (3), respectively. $\alpha$ is hyper-parameter for pseudo-confidence in Definition 2, and $\beta$ is the scaling hyperparameter.

Node-wise inter-channel propagation on $\mathcal{H}^{(i)}$ outputs the final imputed features for $\boldsymbol{X}_{i,:}$. We define node-wise inter-channel propagation as

$$\tilde{\boldsymbol{X}}_{i,:}^{\top} = \hat{\boldsymbol{X}}_{i,:}^{\top} + \boldsymbol{B}^{(i)}(\hat{\boldsymbol{X}}_{i,:} - [m_1, m_2, \cdots, m_F])^{\top}, \qquad (9)$$

where $\tilde{\boldsymbol{X}}_{i,:}$ and $\hat{\boldsymbol{X}}_{i,:}$ are row vectors. Preserving the pre-recovered channel feature values (as self loops), message passing among different channel features is conducted along the directed edges of $\boldsymbol{B}^{(i)}$. After calculating $\tilde{\boldsymbol{X}}_{i,:}$ for $i \in \{1, ..., N\}$, we obtain the final recovered features by concatenating them, i.e., $\tilde{\boldsymbol{X}} = [\tilde{\boldsymbol{X}}_{1,:}^{\top} \ \tilde{\boldsymbol{X}}_{2,:}^{\top} \ \cdots \ \tilde{\boldsymbol{X}}_{N,:}^{\top}]^{\top}$. Moreover, since $\boldsymbol{R}$ is calculated via recovered features $\tilde{\boldsymbol{X}}$ for all nodes in $\mathcal{G}$, channel correlation propagation injects global information into recovered features for $\boldsymbol{X}$. In turn, $\tilde{\boldsymbol{X}}$ is a final output of PC-based feature imputation and is fed to GNN to solve a downstream task.

## 4 EXPERIMENTS

To validate our method, we conducted experiments for two main graph learning tasks: **semi-supervised node classification** and **link prediction**.

### 4.1 EXPERIMENTAL SETUP

**Datasets.** We experimented with six benchmark datasets from two different domains: citation networks (Cora, CiteSeer, PubMed (Sen et al., 2008) and OGBN-Arxiv (Hu et al., 2020)) and recommendation networks (Amazon-Computers and Amazon-Photo (Shchur et al., 2018)). For link

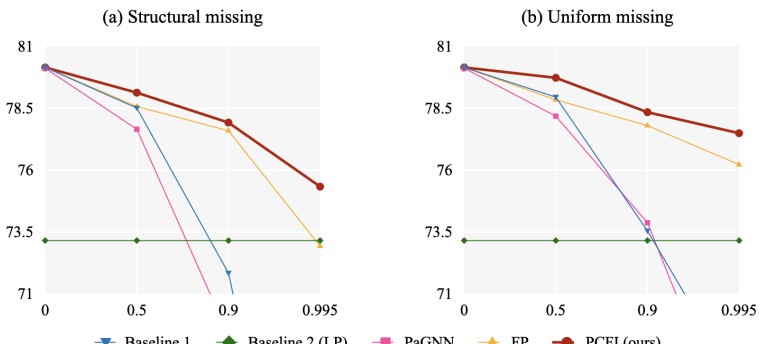

Figure 2: Average accuracy (%) on the six datasets with $r_m \in \{0, 0.5, 0.9, 0.995\}$. sRMGCNN and GCNMF are excepted due to OOM results in certain datasets and the significantly poor performance on all the available datasets, as table 1 shows.

prediction, we evaluated all methods on the five benchmark datasets except OGBN-Arxiv that was caused out of memory. The datasets are described in Appendix A.4.1.

**Compared Methods.** For semi-supervised node classification, we compared our method to two baselines and four state-of-the-art methods. we set **Baseline 1** to a simple scheme that directly fed the graph data with missing features to GNN without recovery, where all missing values in a feature matrix were set to zero. We set **Baseline 2** to label propagation (LP) (Zhu & Ghahramani, 2002) which does not use node features and propagates only partially-known labels for inferring the remaining labels. That is, LP corresponds to the case of 100% feature missing. The four state-of-the-art methods can be categorized into two approaches: *GCN-variant model*={GCNMF (Taguchi et al., 2021), PaGNN (Jiang & Zhang, 2020)} and *feature imputation*= {sRMGCNN (Monti et al., 2017), FP (Rossi et al., 2021)}. While GCN-variant models were designed to perform node classification directly with partially known features, feature imputation methods combine with GNN models for downstream tasks. In Baseline 1, sRMGCNN, FP, and our method, we commonly used vanilla GCN (Kipf & Welling, 2016a) for the downstream task.

For link prediction, we compared our method with sRMGCNN and FP, which are the feature imputation approach. To perform link prediction on the imputed features by each method, graph auto-encoder (GAE) (Kipf & Welling, 2016b) models were adopted. We used features inferred by each method as input of GAE models. We further compared against GCNMF (Taguchi et al., 2021) for link prediction. We report the detailed implementation in Appendix A.3.

**Data Settings.** Regardless of task type, we removed features according to missing rate $r_m$ ($0 < r_m < 1$). Missing features were selected in two ways.

- *Structural missing.* We first randomly selected nodes in a ratio of $r_m$ among all nodes. Then, we assigned all features of the selected nodes to missing (unknown) values (zero).
- *Uniform missing.* We randomly selected features in a ratio of $r_m$ from the node feature matrix $X$, and we set the selected features to missing (unknown) values (zero).

For semi-supervised node classification, we randomly generated 10 different training/validation/test splits, except OGBN-Arxiv where the split was fixed according to the specified criteria. For link prediction, we also randomly generated 10 different training/validation/test splits for each datasets. We describe the generated splits in detail in Appendix A.4.2.

**Hyper-parameters.** Across all the compared methods, we tuned hyper-parameters based on validation set. For PCFI, we analyzed the influence of $\alpha$ and $\beta$ in Appendix A.3.2. We used grid search to find the two hyper-parameters in the range of $0 < \alpha < 1$ and $0 < \beta \leq 1$ on validation sets. For the node classification, $(\alpha, \beta)$ was determined by the best pair from $\{(\alpha, \beta) | \alpha \in \{0.1, 0.2, \cdots, 0.9\}, \beta \in \{10^{-6}, 10^{-5.5}, \cdots, 1\}\}$. For the link prediction, the best $(\alpha, \beta)$ was searched from $\{(\alpha, \beta) | \alpha \in \{0.1, 0.2, \cdots, 0.9\}, \beta \in \{10^{-6}, 10^{-5}, \cdots, 1\}\}$, as shown in Figure 3, 4 of APPENDIX.

**Ablation Study.** We present the ablation study to show the effectiveness of each component (row-stochastic transition matrix, channel-wise inter-node diffusion, and node-wise inter-channel propagation) of PCFI in Appendix A.4.4.

Table 1: Node classification accuracy (%) at missing rate $r_m = 0.995$. OOM denotes out of memory. * denotes incalculable average for six datasets due to OOM results.

| Missing type | Dataset | Baseline 1 | Baseline 2 (LP) | sRMGCNN | GCNMF | PaGNN | FP | PCFI |
|---|---|---|---|---|---|---|---|---|
| Structural missing | Cora | 44.15 ± 8.44 | 74.52 ± 1.60 | 29.31 ± 0.71 | 29.20 ± 1.13 | 30.55 ± 8.85 | 72.84 ± 2.85 | **75.49 ± 2.10** |
| | CiteSeer | 31.68 ± 4.50 | 65.89 ± 2.29 | 24.21 ± 1.35 | 24.50 ± 1.52 | 25.69 ± 3.98 | 59.76 ± 2.47 | **66.18 ± 2.75** |
| | PubMed | 48.20 ± 3.65 | 72.25 ± 3.78 | OOM | 40.19 ± 0.95 | 50.82 ± 4.61 | 72.69 ± 2.66 | **74.66 ± 2.26** |
| | Photo | 79.68 ± 2.17 | 82.42 ± 2.57 | 26.10 ± 1.89 | 26.82 ± 6.33 | 66.91 ± 3.99 | 86.57 ± 1.50 | **87.70 ± 1.29** |
| | Computers | 72.03 ± 1.91 | 76.28 ± 1.43 | 37.15 ± 0.12 | 30.59 ± 9.81 | 56.50 ± 3.29 | 77.45 ± 1.59 | **79.25 ± 1.19** |
| | OGBN-Arxiv | 54.52 ± 0.63 | 67.56 ± 0.00 | OOM | OOM | 57.43 ± 0.36 | 68.23 ± 0.27 | **68.72 ± 0.28** |
| | Average | 55.04 | 73.15 | * | * | 47.98 | 72.92 | **75.33** |
| Uniform missing | Cora | 62.63 ± 2.64 | 74.52 ± 1.60 | 29.32 ± 0.74 | 27.85 ± 2.27 | 53.75 ± 2.03 | 77.55 ± 2.01 | **78.53 ± 1.39** |
| | CiteSeer | 63.19 ± 1.83 | 65.89 ± 2.29 | 24.66 ± 1.90 | 24.29 ± 1.47 | 44.95 ± 2.59 | 68.00 ± 2.16 | **69.40 ± 1.85** |
| | PubMed | 54.70 ± 3.03 | 72.25 ± 3.78 | OOM | 39.47 ± 0.76 | 60.24 ± 3.78 | 73.88 ± 2.35 | **76.44 ± 1.64** |
| | Photo | 85.40 ± 1.33 | 82.42 ± 2.57 | 26.58 ± 1.68 | 25.98 ± 3.90 | 85.30 ± 1.05 | 87.75 ± 1.07 | **88.60 ± 1.30** |
| | Computers | 79.49 ± 1.21 | 76.28 ± 1.43 | 37.16 ± 0.12 | 34.78 ± 4.69 | 78.04 ± 1.18 | 81.47 ± 0.91 | **81.79 ± 0.70** |
| | OGBN-Arxiv | 58.12 ± 0.46 | 67.56 ± 0.00 | OOM | OOM | 65.30 ± 0.22 | 68.67 ± 0.38 | **70.19 ± 0.15** |
| | Average | 67.26 | 73.15 | * | * | 64.6 | 76.22 | **77.49** |

Table 2: Link prediction results (%) at missing rate $r_m = 0.995$. OOM denotes out of memory.

| Dataset | | Full features | Structural missing | | | | Uniform missing | | | |
|---|---|---|---|---|---|---|---|---|---|---|
| | | | sRMGCNN | GCNMF | FP | PCFI | sRMGCNN | GCNMF | FP | PCFI |
| Cora | AP | 92.05 ± 0.75 | 66.34 ± 5.78 | 68.26 ± 1.07 | 83.74 ± 1.05 | **86.45 ± 1.15** | 66.46 ± 5.63 | 67.25 ± 1.10 | 86.31 ± 1.40 | **87.30 ± 1.33** |
| | AUC | 92.58 ± 0.86 | 68.80 ± 6.44 | 71.09 ± 0.87 | 86.12 ± 1.04 | **88.26 ± 0.97** | 68.87 ± 6.36 | 70.78 ± 0.86 | 88.73 ± 1.16 | **89.24 ± 1.08** |
| CiteSeer | AP | 90.50 ± 0.92 | 67.75 ± 1.95 | 67.75 ± 1.98 | 79.74 ± 1.71 | **80.12 ± 1.59** | 64.35 ± 5.19 | 65.71 ± 1.80 | 82.02 ± 1.95 | **82.98 ± 2.30** |
| | AUC | 91.65 ± 0.99 | 69.08 ± 1.88 | 69.10 ± 1.95 | 83.24 ± 1.43 | **83.88 ± 1.30** | 66.30 ± 5.65 | 68.55 ± 1.72 | 85.81 ± 1.47 | **86.28 ± 1.77** |
| PubMed | AP | 95.82 ± 0.27 | OOM | **87.14 ± 0.28** | 78.93 ± 1.51 | 82.65 ± 0.91 | OOM | 81.67 ± 2.27 | 77.05 ± 3.54 | **85.26 ± 0.36** |
| | AUC | 95.95 ± 0.26 | OOM | 86.07 ± 0.31 | 84.30 ± 0.98 | **87.02 ± 0.41** | OOM | 82.70 ± 1.39 | 83.26 ± 2.24 | **88.52 ± 0.20** |
| Photo | AP | 95.76 ± 0.38 | 81.48 ± 0.29 | 81.45 ± 0.30 | 94.05 ± 1.18 | **96.40 ± 0.42** | 81.53 ± 0.27 | 81.48 ± 0.30 | 95.97 ± 0.21 | **97.07 ± 0.21** |
| | AUC | 95.34 ± 0.42 | 81.07 ± 0.33 | 81.03 ± 0.34 | 93.57 ± 1.06 | **96.01 ± 0.49** | 81.14 ± 0.29 | 81.07 ± 0.33 | 95.54 ± 0.24 | **96.89 ± 0.23** |
| Computers | AP | 93.78 ± 1.16 | 83.37 ± 0.17 | 83.33 ± 0.17 | 90.57 ± 1.23 | **94.65 ± 0.40** | 83.39 ± 0.18 | 83.36 ± 0.17 | 93.96 ± 0.24 | **95.98 ± 0.21** |
| | AUC | 93.79 ± 1.09 | 83.66 ± 0.24 | 83.62 ± 0.24 | 90.92 ± 1.05 | **94.67 ± 0.43** | 83.68 ± 0.26 | 83.65 ± 0.25 | 93.90 ± 0.24 | **96.03 ± 0.22** |

## 4.2 SEMI-SUPERVISED NODE CLASSIFICATION RESULTS

Figure 2 demonstrates the trend of an average accuracy of compared methods for node classification on six datasets with different $r_m$. The performance gain of PCFI is remarkable at $r_m = 0.995$. In contrast, the average accuracy of existing methods rapidly decrease as $r_m$ increases and are overtaken by LP which does not utilize features. In the case of uniform missing features, FP exhibits better resistance than LP, but the gap from ours increases as $r_m$ increases.

Table 1 illustrates the detailed results of node classification with $r_m = 0.995$. sRMGCNN and GCNMF show significantly low performance for all experiments in this extremely challenging environment. Baseline 2 (LP) outperforms PaGNN in general, and even FP shows worse accuracy than Baseline 2 (LP) in certain settings. For all the datasets, PCFI performed in a manner that was superior to the other methods at $r_m = 0.995$.

## 4.3 LINK PREDICTION RESULTS

Table 2 demonstrates the results for the link prediction task at $r_m$=0.995. PCFI achieves state-of-the-art performance across all settings except PubMed with structural missing. Based on the results on semi-supervised node classification and link prediction, which are representative graph learning tasks, PCFI shows the effectiveness at a very high rate of missing features.

## 5 CONCLUSION

We introduced a novel concept of channel-wise confidence to impute highly rated missing features in a graph. To replace the unavailable true confidence, we designed a pseudo-confidence obtainable from the shortest path distance of each channel feature on a node. Using the pseudo-confidence, we developed a new framework for missing feature imputation that consists of channel-wise inter-node diffusion and node-wise inter-channel propagation. As validated in experiments, the proposed method demonstrates outperforming performance on both node classification and link prediction. The channel-wise confidence approach for missing feature imputation can be straightforwardly applied to various graph-related downstream tasks with missing node features.

## ETHICS STATEMENT

The intentionally removed private or confidential information can be recovered using the proposed method and the recovered information can be misused. Therefore, the work is suggested to be used for positive impacts on society in areas such as health care (Wang et al., 2020; Deng et al., 2020), crime prediction (Wang et al., 2021), and weather forecasting (Han et al., 2022).

## REPRODUCIBILITY STATEMENT

For theoretical results, we explained the assumptions and the complete proofs of all theoretical results in Section 3.4, 3.5, and Appendix. In addition, we include the data and implementation details to reproduce the experimental results in Section 4 and Appendix A.3. The codes are available at https://github.com/daehoum1/pcfi.

## ACKNOWLEDGMENTS

This work was supported by IITP grant funded by Korea government(MSIT) [No.B0101-15-0266, Development of High Performance Visual BigData Discovery Platform for Large-Scale Realtime Data Analysis; NO.2021-0-01343, Artificial Intelligence Graduate School Program (Seoul National University)]

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

# A  APPENDIX

## A.1  PROOF OF PROPOSITION 1

**Proposition 1.** *If $\boldsymbol{S}_{i,d} = m \geq 1$, $v_i$ is a missing node, then $\xi_{j/i,d}$ for $v_j \in \mathcal{N}(v_i)$ is given by*

$$
\begin{aligned}
\xi_{j/i,d} &= \alpha^{-1} \ if \ \boldsymbol{S}_{i,d} > \boldsymbol{S}_{j,d}, \\
\xi_{j/i,d} &= 1 \ if \ \boldsymbol{S}_{i,d} = \boldsymbol{S}_{j,d}, \\
\xi_{j/i,d} &= \alpha \ if \ \boldsymbol{S}_{i,d} < \boldsymbol{S}_{j,d},
\end{aligned}
$$

*Otherwise, $v_i$ is a source node ($\boldsymbol{S}_{i,d} = 0$), then $\xi_{j/i,d}$ for $v_j \in \mathcal{N}(v_i)$ is given by*

$$
\begin{aligned}
\xi_{j/i,d} &= 1 \ if \ v_j \ is \ a \ source \ node \ (\boldsymbol{S}_{j,d} = 1), \\
\xi_{j/i,d} &= \alpha \ if \ v_j \ is \ a \ missing \ node \ (\boldsymbol{S}_{j,d} = 0).
\end{aligned}
$$

*Proof.* Let $v_a$ and $v_b$ be arbitrary nodes, and let $\delta(v_a, v_b)$ denote the number of edges in the shortest path between $v_a$ and $v_b$. The shortest path distance from $v_i$ to its nearest source node for the $d$-th feature channel, $\boldsymbol{S}_{i,d}$, is given by

$$
\boldsymbol{S}_{i,d} = \min\{\delta(v_i, v_s) | \, v_s \in \mathcal{V}_k^{(d)}\}.
$$

*Claim 1*: $\boldsymbol{S}_{i,d} = 0 \Leftrightarrow v_i \in \mathcal{V}_k^{(d)}$. *Proof*: Since $v_i$ is a source node, $\boldsymbol{S}_{i,d} = 0$.

*Claim 2*: $\boldsymbol{S}_{i,d} \geq 1 \Leftrightarrow v_i \notin \mathcal{V}_k^{(d)} \Leftrightarrow v_i \in \mathcal{V}_u^{(d)}$. *Proof*: Since $v_i$ is not a known node ($v_i \notin \mathcal{V}_k^{(d)}$) if and only of $v_i$ is unknown node ($v_i \in \mathcal{V}_u^{(d)}$), then $\boldsymbol{S}_{i,d} \geq 1$ is obvious.

Let $v_s$ be a known node such that $\delta(v_s, v_i) = m$ which exists because $\boldsymbol{S}_{i,d} = m$. Then $\delta(v_s, v_j) \leq \delta(v_s, v_i) + \delta(v_i, v_j) = m + 1$ holds by the triangle inequality since shortest path distance is a metric on the graph. This proves that $\boldsymbol{S}_{j,d} \leq m + 1$, and also included the case of $S_{i,d} = 0$ as a special case.

Assume that there is some known node $v_{s'}$ such that $\delta(v_j, v_{s'}) \leq m - 2$. Then $\delta(v_i, v_{s'}) \leq \delta(v_i, v_j) + \delta(v_j, v_{s'}) \leq 1 + m - 2 = m - 1$ by the triangle inequality. However, this contradicts $\boldsymbol{S}_{i,d} = m$. Therefore, for all source node $s'$, $\delta(v_j, v_{s'}) \geq m - 1$ which implies $\boldsymbol{S}_{j,d} \geq m - 1$..

Then, the following *Claim 3* also holds.

*Claim 3*: If $\boldsymbol{S}_{i,d} = m \geq 1$, then $\boldsymbol{S}_{j,d} - \boldsymbol{S}_{i,d} \in \{-1, 0, 1\}$ for $v_j \in \mathcal{N}(v_i)$. Otherwise, if $\boldsymbol{S}_{i,d} = 0$, then $\boldsymbol{S}_{j,d} - \boldsymbol{S}_{i,d} \in \{0, 1\}$ for $v_j \in \mathcal{N}(v_i)$

According to *Claim 3* and $\xi_{j/i,d} = \alpha^{\boldsymbol{S}_{j,d} - \boldsymbol{S}_{i,d}}$ in Definition 3 of the main text, the proposition 1 holds trivially.

$\square$

## A.2 CONVERGENCE OF CHANNEL-WISE INTER-NODE DIFFUSION

The convergence of the proposed Channel-wise Inter-node Diffusion is presented in the following Proposition.

**Proposition A.1.** *The channel-wise inter-node diffusion matrix for the $d$-th channel, $\widehat{\boldsymbol{W}}^{(d)}$, is expressed by*

$$\widehat{\boldsymbol{W}}^{(d)} = \begin{bmatrix} \boldsymbol{I} & \boldsymbol{0}_{ku} \\ \overline{\boldsymbol{W}}_{uk}^{(d)} & \overline{\boldsymbol{W}}_{uu}^{(d)} \end{bmatrix},$$

*where $\overline{\boldsymbol{W}}^{(d)}$ is the row-stochastic matrix calculated by normalizing $\boldsymbol{W}^{(d)}$. The recursion in channel-wise inter-node diffusion for the $d$-th channel is defined by*

$$\hat{\boldsymbol{x}}^{(d)}(0) = \begin{bmatrix} \boldsymbol{x}_k^{(d)} \\ \boldsymbol{0}_u \end{bmatrix}$$

$$\hat{\boldsymbol{x}}^{(d)}(t) = \widehat{\boldsymbol{W}}^{(d)} \hat{\boldsymbol{x}}^{(d)}(t-1)$$

*Then, $\lim_{K\to\infty} \hat{\boldsymbol{x}}_u^{(d)}(K)$ converges to $(\boldsymbol{I} - \overline{\boldsymbol{W}}_{uu}^{(d)})^{-1}\overline{\boldsymbol{W}}_{uk}^{(d)}\boldsymbol{x}_k^{(d)}$, where $\boldsymbol{x}_k^{(d)}$ is the known feature of the $d$-th channel.*

The proof of this Proposition follows that of (Rossi et al., 2021) which proves the case of a symmetrically-normalized diffusion matrix. In our proof, the diffusion matrix is not symmetric. For proof of Proposition A.1, we first give Lemma A.1 and A.2.

**Lemma A.1.** $\overline{\boldsymbol{W}}^{(d)}$ *is the row-stochastic matrix calculated by normalizing $\boldsymbol{W}^{(d)}$ which is the weighted adjacency matrix of the connected graph $\mathcal{G}$. That is, $\overline{\boldsymbol{W}}^{(d)} = (\boldsymbol{D}^{(d)})^{-1}\boldsymbol{W}^{(d)}$ where $\boldsymbol{D}_{ii}^{(d)} = \sum_j \boldsymbol{W}_{i,j}$. Let $\overline{\boldsymbol{W}}_{uu}^{(d)}$ be the $|\hat{\boldsymbol{x}}_u^{(d)}| \times |\hat{\boldsymbol{x}}_u^{(d)}|$ bottom-right submatrix of $\overline{\boldsymbol{W}}^{(d)}$, and let $\rho(\cdot)$ denote spectral radius. Then, $\rho(\overline{\boldsymbol{W}}_{uu}^{(d)}) < 1$.*

*Proof.* Let $\overline{\boldsymbol{W}}_{uu0}^{(d)} \in \mathbb{R}^{N\times N}$ be the matrix where the bottom right submatrix is $\overline{\boldsymbol{W}}_{uu}^{(d)}$ and all the other elements are zero. That is,

$$\overline{\boldsymbol{W}}_{uu0}^{(d)} = \begin{bmatrix} \boldsymbol{0}_{kk} & \boldsymbol{0}_{ku} \\ \boldsymbol{0}_{uk} & \overline{\boldsymbol{W}}_{uu}^{(d)} \end{bmatrix}$$

where $\boldsymbol{0}_{kk} \in \{0\}^{|\hat{\boldsymbol{x}}_k^{(d)}|\times|\hat{\boldsymbol{x}}_k^{(d)}|}$, $\boldsymbol{0}_{ku} \in \{0\}^{|\hat{\boldsymbol{x}}_k^{(d)}|\times|\hat{\boldsymbol{x}}_u^{(d)}|}$, and $\boldsymbol{0}_{uk} \in \{0\}^{|\hat{\boldsymbol{x}}_u^{(d)}|\times|\hat{\boldsymbol{x}}_k^{(d)}|}$.

Since $\overline{\boldsymbol{W}}^{(d)}$ is the weighted adjacency matrix of connected graph $\mathcal{G}$, $\overline{\boldsymbol{W}}_{uu0}^{(d)} \leq \overline{\boldsymbol{W}}^{(d)}$ element-wisely and $\overline{\boldsymbol{W}}_{uu0}^{(d)} \neq \overline{\boldsymbol{W}}^{(d)}$. Moreover, since $\overline{\boldsymbol{W}}_{uu0}^{(d)} + \overline{\boldsymbol{W}}^{(d)}$ is a weighted adjacency matrix of a strongly connected graph, $\overline{\boldsymbol{W}}_{uu0}^{(d)} + \overline{\boldsymbol{W}}^{(d)}$ is irreducible due to Theorem 2.2.7 of (Berman & Plemmons, 1994). Then, by Corollary 2.1.5 of (Berman & Plemmons, 1994), $\rho(\overline{\boldsymbol{W}}_{uu0}^{(d)}) < \rho(\overline{\boldsymbol{W}}^{(d)})$. Since the spectral radius of a stochastic matrix is one (Theorem 2.5.3 in (Berman & Plemmons, 1994)), $\rho(\overline{\boldsymbol{W}}^{(d)}) = 1$. Furthermore, since $\overline{\boldsymbol{W}}_{uu0}^{(d)}$ and $\overline{\boldsymbol{W}}_{uu}^{(d)}$ share the same non-zero eigenvalues, $\rho(\overline{\boldsymbol{W}}_{uu}^{(d)}) = \rho(\overline{\boldsymbol{W}}_{uu}^{(d)})$. Finally, $\rho(\overline{\boldsymbol{W}}_{uu}^{(d)}) = \rho(\overline{\boldsymbol{W}}_{uu0}^{(d)}) < \rho(\overline{\boldsymbol{W}}^{(d)}) = 1$. $\square$

**Lemma A.2.** $\boldsymbol{I} - \overline{\boldsymbol{W}}_{uu}^{(d)}$ *is invertible where $\boldsymbol{I}$ is the $|\hat{\boldsymbol{x}}_u^{(d)}| \times |\hat{\boldsymbol{x}}_u^{(d)}|$ identity matrix.*

*Proof.* Since 1 is not an eigenvalue of $\overline{\boldsymbol{W}}_{uu}^{(d)}$ by Lemma A.1, 0 is not an eigenvlaue of $\boldsymbol{I} - \overline{\boldsymbol{W}}_{uu}^{(d)}$. Thus $\boldsymbol{I} - \overline{\boldsymbol{W}}_{uu}^{(d)}$ is invertible. $\square$

In the following, we give the proof of Proposition 1.
*Proof of Proposition 1.* Unfolding the recurrence relation gives us

$$\hat{\boldsymbol{x}}^{(d)}(t) = \begin{bmatrix} \hat{\boldsymbol{x}}_k^{(d)}(t) \\ \hat{\boldsymbol{x}}_u^{(d)}(t) \end{bmatrix} = \begin{bmatrix} \boldsymbol{I} & \boldsymbol{0}_{ku} \\ \overline{\boldsymbol{W}}_{uk}^{(d)} & \overline{\boldsymbol{W}}_{uu}^{(d)} \end{bmatrix} \begin{bmatrix} \hat{\boldsymbol{x}}_k^{(d)}(t-1) \\ \hat{\boldsymbol{x}}_u^{(d)}(t-1) \end{bmatrix} = \begin{bmatrix} \hat{\boldsymbol{x}}_k^{(d)}(t-1) \\ \overline{\boldsymbol{W}}_{uk}^{(d)}\hat{\boldsymbol{x}}_k^{(d)}(t-1) + \overline{\boldsymbol{W}}_{uu}^{(d)}\hat{\boldsymbol{x}}_u^{(d)}(t-1) \end{bmatrix}.$$

Since $\hat{\boldsymbol{x}}_k^{(d)}(t) = \hat{\boldsymbol{x}}_k^{(d)}(t-1)$ in the first $|\hat{\boldsymbol{x}}_k^{(d)}|$ rows, $\hat{\boldsymbol{x}}_k^{(d)}(K) = ... = \hat{\boldsymbol{x}}_k^{(d)}$. That is, $\hat{\boldsymbol{x}}_k^{(d)}(K)$ remains $\boldsymbol{x}_k^{(d)}$. Hence $\lim\limits_{K \to \infty} \hat{\boldsymbol{x}}_k^{(d)}(K)$ converges to $\boldsymbol{x}_k^{(d)}$.

Now, we just consider the convergence of $\lim\limits_{K \to \infty} \hat{\boldsymbol{x}}_u^{(d)}(K)$. Unrolling the recursion of the last $|\hat{\boldsymbol{x}}_u^{(d)}|$ rows become,

$$
\begin{aligned}
\hat{\boldsymbol{x}}_u^{(d)}(K) &= \overline{\boldsymbol{W}}_{uk}^{(d)} \boldsymbol{x}_k^{(d)} + \overline{\boldsymbol{W}}_{uu}^{(d)} \hat{\boldsymbol{x}}_u^{(d)}(K-1) \\
&= \overline{\boldsymbol{W}}_{uk}^{(d)} \boldsymbol{x}_k^{(d)} + \overline{\boldsymbol{W}}_{uu}^{(d)} (\overline{\boldsymbol{W}}_{uk}^{(d)} \boldsymbol{x}_k^{(d)} + \overline{\boldsymbol{W}}_{uu}^{(d)} \hat{\boldsymbol{x}}_u^{(d)}(K-2)) \\
&= ... \\
&= \Big( \sum_{t=0}^{K-1} (\overline{\boldsymbol{W}}_{uu}^{(d)})^t \Big) \overline{\boldsymbol{W}}_{uk}^{(d)} \boldsymbol{x}_k^{(d)} + (\overline{\boldsymbol{W}}_{uu}^{(d)})^K \hat{\boldsymbol{x}}_u^{(d)}(0)
\end{aligned}
$$

Since $\lim\limits_{K \to \infty} (\overline{\boldsymbol{W}}_{uu}^{(d)})^K = 0$ by Lemma A.1, $\lim\limits_{K \to \infty} (\overline{\boldsymbol{W}}_{uu}^{(d)})^K \hat{\boldsymbol{x}}_u^{(d)}(0) = 0$ regardless of the initial state for $\hat{\boldsymbol{x}}_u^{(d)}(0)$. (We replace $\hat{\boldsymbol{x}}_u^{(d)}(0)$ with a zero column vector for simplicity.) Thus, it remains to consider $\lim\limits_{K \to \infty} (\sum_{t=0}^{K-1} (\overline{\boldsymbol{W}}_{uu}^{(d)})^t) \overline{\boldsymbol{W}}_{uk}^{(d)} \boldsymbol{x}_k^{(d)}$.

Since $\rho(\overline{\boldsymbol{W}}_{uu}^{(d)}) < 1$ by Lemma A.1 and $(\boldsymbol{I} - \overline{\boldsymbol{W}}_{uu}^{(d)})^{-1}$ is invertible by Lemma A.2, the geometric series converges as follows

$$
\lim_{K \to \infty} \hat{\boldsymbol{x}}_u^{(d)}(K) = \lim_{K \to \infty} \Big( \sum_{t=0}^{K-1} (\overline{\boldsymbol{W}}_{uu}^{(d)})^t \Big) \overline{\boldsymbol{W}}_{uk}^{(d)} \boldsymbol{x}_k^{(d)} = (\boldsymbol{I} - \overline{\boldsymbol{W}}_{uu}^{(d)})^{-1} \overline{\boldsymbol{W}}_{uk}^{(d)} \boldsymbol{x}_k^{(d)}.
$$

Thus, the recursion in channel-wise inter-node diffusion converges. $\qquad\square$

A.3 IMPLEMENTATION

A.3.1 IMPLEMENTATION DETAILS

We used Pytorch (Paszke et al., 2017) and Pytorch Geometric (Fey & Lenssen, 2019) for the experiments on an NVIDIA GTX 2080 Ti GPU with 11GB of memory.

**Node classification.** We trained GCN-variant models (GCNMF, PaGNN) and GCN models for feature imputation methods (Baseline 1, sRMGCNN, FP, PCFI) as follows. We used Adam optimizer (Kingma & Ba, 2014) and set the maximal number of epochs to 10000. We used an early stopping strategy with patience of 200 epochs. By grid search on each validation set, learning rates of all experiments are chosen from $\{0.01, 0.005, 0.001, 0.0001\}$, and dropout (Srivastava et al., 2014) was applied with $p$ selected in $\{0.0, 0.25, 0.5\}$.

**Link prediction.** For GCNMF and GAE used as common downstream models for feature imputation methods, we trained the models with Adam optimizer for 200 iterations. By grid search on the validation set, learning rates of all methods are searched from $\{0.1, 0.01, 0.005, 0.001, 0.0001\}$ for each dataset, and dropout was applied to each layer with $p$ searched from $\{0.0, 0.25, 0.5\}$. As specified in (Kipf & Welling, 2016b) and (Taguchi et al., 2021), we used a 32-dim hidden layer and 16-dim latent variables for the all auto-encoder models.

For all the compared methods, we followed all the hyper-parameters in original papers or codes if feasible. If hyper-parameters (the number of layers and hidden dimension) of a model for certain datasets are not clarified in the papers, we searched the hyper-parameters using grid search. In that case, we searched the number of layers from $\{2, 3\}$ and the hidden dimension from $\{16, 32, 64, 128, 256\}$.

We present the pseudo-code of our PCFI in Sec. A.6. Our code will be available upon publication.

A.3.2 PCFI HYPER-PARAMETERS

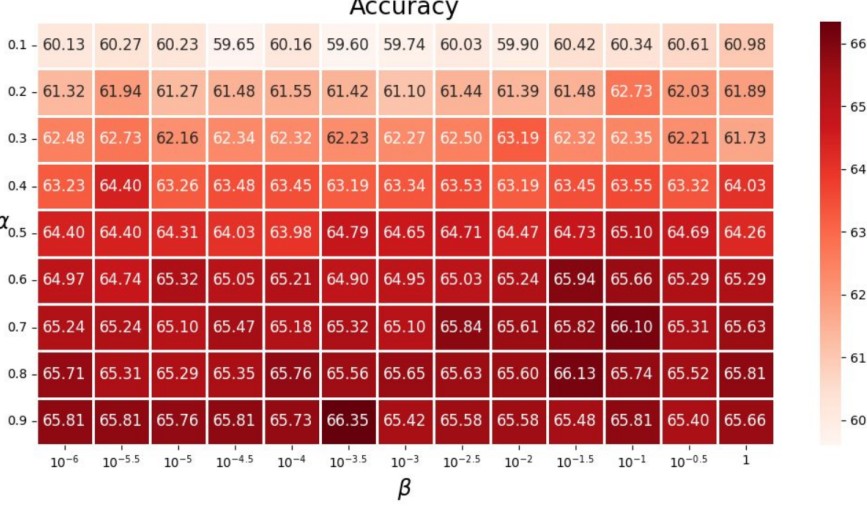

Figure 3: Node classification accuracy on CiteSeer with different $\alpha$ and $\beta$. The experiments are conducted under a structural-missing setting with $r_m = 0.995$.

We set $K$ to 100 throughout all the experiments. PCFI has two hyper-parameters $\alpha$ and $\beta$. $\alpha$ is used to calculate PC for channel-wise inter-node diffusion and node-wise inter-channel propagation. $\beta$ controls the degree of node-wise inter-channel propagation. To analyze the effects of the hyper-parameter, we conducted experiments with various $\alpha$ and $\beta$. Figure 3 and Figure 4 demonstrate the influence of $\alpha$ and $\beta$.

For node classification, we set search range of $\alpha$ and $\beta$ same as in Figure 3. For node classification, we chose $\alpha$ from $\{0.1, 0.2, ..., 0.9\}$, and $\beta$ from $\{10^{-6}, 10^{-5.5}, 10^{-5}, ..., , 1\}$ using a validation set. Then, for link prediction, we set search range of $\alpha$ and $\beta$ same as in Figure 4. We selected $\alpha$ from

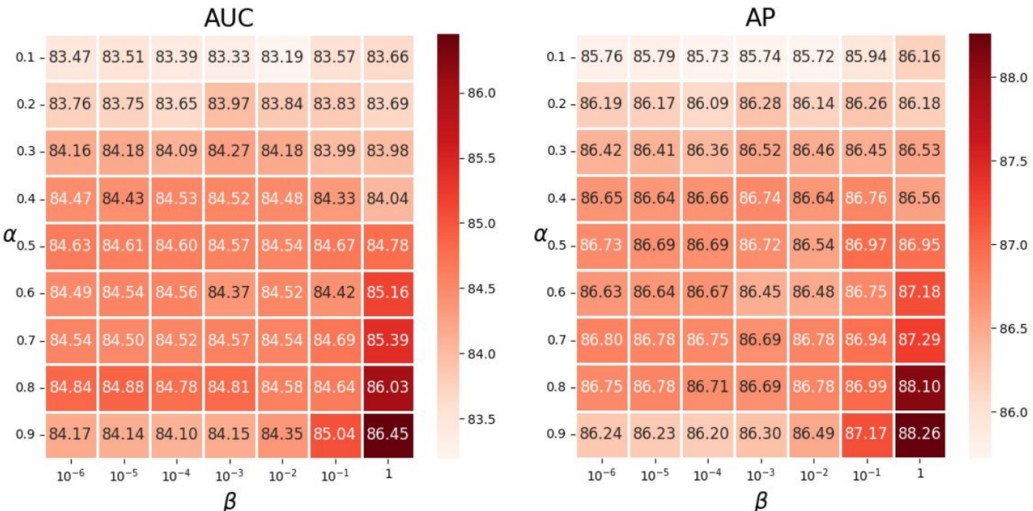

Figure 4: Link prediction results on Cora with different $\alpha$ and $\beta$. The experiments are conducted under a structural-missing setting with $r_m = 0.995$.

Table 3: Hyper-parameters of PCFI used in node classification.

| Missing type | | Structural missing | | | Uniform missing | | |
|---|---|---|---|---|---|---|---|
| $r_m$ | | 0.995 | 0.9 | 0.5 | 0.995 | 0.9 | 0.5 |
| Cora | $\alpha$ | 0.9 | 0.9 | 0.6 | 0.7 | 0.5 | 0.5 |
| | $\beta$ | 1 | $10^{-1}$ | 1 | $10^{-2}$ | $10^{-1}$ | 1 |
| CiteSeer | $\alpha$ | 0.8 | 0.6 | 0.5 | 0.8 | 0.5 | 0.1 |
| | $\beta$ | $10^{-1.5}$ | $10^{-0.5}$ | 1 | $10^{-0.5}$ | 1 | 1 |
| PubMed | $\alpha$ | 0.8 | 0.9 | 0.9 | 0.7 | 0.8 | 0.2 |
| | $\beta$ | $10^{-1}$ | $10^{-0.5}$ | $10^{-0.5}$ | $10^{-2.5}$ | 1 | 1 |
| Photo | $\alpha$ | 0.2 | 0.4 | 0.6 | 0.2 | 0.2 | 0.5 |
| | $\beta$ | $10^{-4}$ | $10^{-6}$ | $10^{-2.5}$ | $10^{-1.5}$ | $10^{-6}$ | $10^{4.5}$ |
| Computers | $\alpha$ | 0.1 | 0.1 | 0.3 | 0.1 | 0.2 | 0.4 |
| | $\beta$ | $10^{-3.5}$ | $10^{-4}$ | $10^{-5.5}$ | $10^{-2.5}$ | $10^{-4}$ | $10^{-5.5}$ |
| OGBN-Arxiv | $\alpha$ | 0.1 | 0.4 | 0.2 | 0.2 | 0.8 | 0.8 |
| | $\beta$ | $10^{-6}$ | $10^{-6}$ | $10^{-6}$ | $10^{-4}$ | $10^{-6}$ | $10^{-2.5}$ |

Table 4: Hyper-parameters of PCFI used in link prediction.

| missing type | Dataset | Cora | CiteSeer | PubMed | Photo | Computers |
|---|---|---|---|---|---|---|
| Structural | $\alpha$ | 0.9 | 0.9 | 0.2 | 0.1 | 0.1 |
| | $\beta$ | 1 | $10^{-1}$ | 1 | $10^{-1}$ | $10^{-1}$ |
| Uniform | $\alpha$ | 0.9 | 0.6 | 0.3 | 0.1 | 0.1 |
| | $\beta$ | 1 | 1 | 1 | 1 | 1 |

$\{0.1, 0.2, ..., 0.9\}$, and $\beta$ from $\{10^{-6}, 10^{-5}, 10^{-4}, ..., , 1\}$. To find the best hyper-parameter, we used grid search on a validation set. The detailed setting of hyper-parameters for all datasets used in our paper are listed in Table 3 and Table 4.

To train downstream GCNs added to PCFI for node classification, we fix a learning rate to 0.005. Then, to train downstream GAE added to PCFI for link prediction, we set a learning rate to 0.01 and 0.001 for {Cora, CiteSeer, PubMed} and {Photo, Computers}, respectively.

**Downstream GCN for node classification.** We set the number of layers to 3, and We fix dropout rate $p = 0.5$. The hidden dimension was set to 64 for all datasets except OGBN-Arxiv where 256 is used. For OGBN-Arxiv, as the Jumping Knowledge scheme (Xu et al., 2018) with max aggregation was applied to FP, we also utilized the scheme.

### A.3.3 IMPLEMENTATION OF BASELINES

**GCNMF** (Taguchi et al., 2021)**.** We used publicly released code by the authors. The code for GCNMF[1] is MIT licensed.

**FP** (Rossi et al., 2021)**.** We used publicly released code by the authors. The code for FP[2] is Apache-2.0 licensed.

**PaGNN** (Jiang & Zhang, 2020)**.** We used re-implemented Apache-2.0-licensed code[3] since we could not find officially released code by the authors for PaGNN.

**sRMGCNN** (Monti et al., 2017)**.** Due to the compatibility problem from the old-version Tensor-flow (Abadi et al., 2016) of the code, we only updated the version of publicly released code[4] to Tensorflow 2.3.0. The code is GPL-3.0 licensed.

**Label propagation** (Zhu & Ghahramani, 2002)**.** We employed re-implemented code included in MIT-licensed Pytorch Geometric. We tuned hyper-parameter $\alpha$ of LP in $\{0.1, 0.2, 0.3, 0.4, 0.5, 0.6, 0.7, 0.8, 0.9, 0.95\}$ by grid search.

## A.4 EXPERIMENTS

### A.4.1 DATASETS

All the datasets used in our experiments are publicly available from the MIT-licensed Pytorch Geometric package. We conducted all the experiments on the largest connected component of each given graph. For a disconnected graph, we can simply apply PCFI to each connected component independently. The description for the datasets is summarized in Table 5.

Table 5: Dataset statistics.

| Dataset | # Nodes | # Edges | # Features | # Classes |
|---|---|---|---|---|
| Cora | 2,485 | 5,069 | 1,433 | 7 |
| CiteSeer | 2,120 | 3,679 | 3,703 | 6 |
| PubMed | 19,717 | 44,324 | 500 | 3 |
| Photo | 7,487 | 119,043 | 745 | 8 |
| Computers | 13,381 | 245,778 | 767 | 10 |
| OGBN-Arxiv | 169,343 | 1,166,243 | 128 | 40 |

---

[1]https://github.com/marblet/GCNmf

[2]https://github.com/twitter-research/feature-propagation

[3]https://github.com/twitter-research/feature-propagation

[4]https://github.com/fmonti/mgcnn

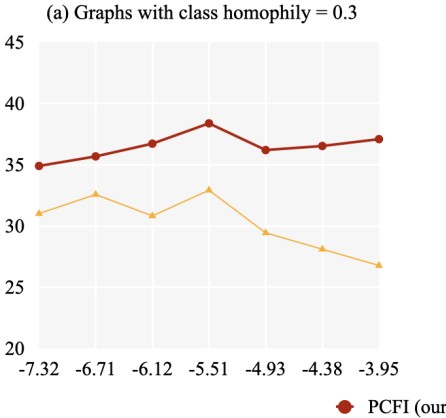 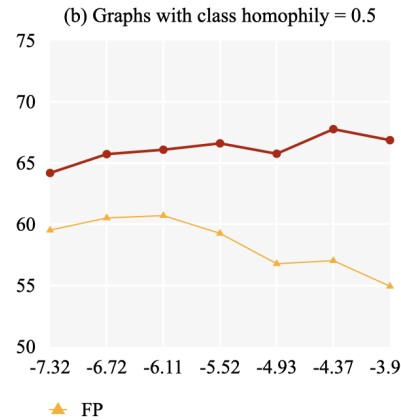

Figure 5: Node classification accuracy (%) on the synthetic graphs according to $-\log(E_d)$, where $E_d$ is the Dirichlet energy (An increase in $E_d$ means an increase in homophily.). The experiments are conducted under a structural-missing setting with $r_m = 0.995$. The proposed PCFI consistently outperforms FP and the performance gap widens as homophily increases.

### A.4.2 RANDOM SPLIT GENERATION

**Node classification.** We randomly generated 10 different training/validation/test splits, except OGBN-Arxiv where the split was given according to the published year of papers. As the setting in (Klicpera et al., 2019), in each generated split, 20 nodes per class were assigned to the training nodes. Then, the number of validation nodes is determined by the number that becomes 1500 by adding to the number of the training nodes. As the test nodes, we used all nodes except training and validation nodes.

**Link prediction.** We randomly generated 10 different training/validation/test splits for each datasets. In each split, we applied the same split ratio regardless of datasets. As the setting in (Kipf & Welling, 2016b), we commonly used 85% edges for train, 5% edges for validation, and the 10% edges for test.

### A.4.3 GAIN OF PCFI ACCORDING TO FEATURE HOMOPHILY

In this section, since pseudo-confidence is based on the assumption that linked nodes have similar features, we explored how feature homophily of graphs impacts the performance of PCFI. We further analyzed the gain of PCFI over FP in terms of feature homophily. For the experiments, we generated features with different feature homophily on graphs from the synthetic dataset where each graph contains 5000 nodes that belong to one of 10 classes.

We selected two graphs (graphs with class homophily of 0.3 and 0.5) from the dataset. Preserving the graph structure and class distribution, we newly generated multiple sets of node features for each graph so that each generated graph has different feature homophily. The features for nodes were sampled from overlapping 10 5-dimensional Gaussians which correspond to each class and the means of the Gaussians are set differently to be the same distance from each other. For the covariance matrix of each Gaussian, diagonal elements were set to the same value and the other elements were set to 0.1 times the diagonal element. That is, the covariance between different channels was set to 0.1 times the variance of a channel. For each feature generation of a graph, we changed the scale of the covariance matrix so that features are created with different feature homophily. As the scale of the covariance matrix decreases, the overlapped area between Gaussians decreases. By doing so, more similar features are generated within the same class and the graph has higher feature homophily. Then, for semi-supervised node classification, we randomly generated 10 different training/validation/test splits. For each split, we set the numbers of nodes in training, validation, and test set to be equal.

Figure 5 demonstrates the trend of the accuracy of PCFI and FP under a structural-missing setting with $r_m = 0.995$. We can confirm that the gain of PCFI over FP exceeds 10% on both graphs with high feature homophily. Furthermore, PCFI shows superior performance regardless of levels of feature homophily. This is because confidence based on feature homophily is the valid concept on graphs with missing features and pseudo-confidence is designed properly to replace confidence.

### A.4.4 ABLATION STUDY

To verify the effectiveness of each element of PCFI, we carried out an ablation study. We started by measuring the performance of FP that performs simple graph diffusion with a symmetrically-normalized transition matrix. Firstly, we changed the normalization type of the transition matrix. We replaced the symmetrically-normalized transition matrix with a row-stochastic transition matrix (row-ST in Table 6 and Table7). The row-stochastic transition matrix leads to feature recovery at the same scale of actual features. Secondly, we introduced PC to the diffusion process, which means PC-based channel-wise inter-node diffusion was performed (CID in Table 6 and Table7). Lastly, we performed node-wise channel propagation on recovered features obtained by channel-wise inter-node diffusion (NIP in Table 6 and Table7). We compared the performance of these four cases. In this experiment, for CiteSeer, we performed node classification under structural-missing setting with $r_m = 0.995$. Then, for Cora and PubMed, we performed link prediction with $r_m = 0.995$. We applied structural missing and uniform missing to Cora and PubMed, respectively. As shown in Table 6 and Table 7, each component of PCFI contributes to performance improvement throughout the various settings.

Table 6: Ablation study of PCFI. row-ST, CID and, NIP denote a row-stochastic transition matrix, channel-wise inter-node diffusion, and node-wise inter-channel propagation, respectively.

| row-ST | CID | NIP | CiteSeer |
|:---:|:---:|:---:|:---:|
| ✗ | ✗ | ✗ | $59.76 \pm 2.47$ |
| ✓ | ✗ | ✗ | $64.80 \pm 2.60$ |
| ✓ | ✓ | ✗ | $65.40 \pm 2.77$ |
| ✓ | ✓ | ✓ | $\mathbf{66.18 \pm 2.75}$ |

Table 7: Ablation study of PCFI. row-ST, CID and, NIP denote a row-stochastic transition matrix, channel-wise inter-node diffusion, and node-wise inter-channel propagation, respectively.

| row-ST | CID | NIP | Cora | | PubMed | |
|:---:|:---:|:---:|:---:|:---:|:---:|:---:|
| | | | AUC (%) | AP (%) | AUC (%) | AP (%) |
| ✗ | ✗ | ✗ | $83.74 \pm 1.05$ | $86.12 \pm 1.04$ | $77.05 \pm 3.54$ | $83.26 \pm 2.24$ |
| ✓ | ✗ | ✗ | $83.96 \pm 1.02$ | $86.14 \pm 1.07$ | $80.72 \pm 1.28$ | $84.99 \pm 0.68$ |
| ✓ | ✓ | ✗ | $84.16 \pm 1.23$ | $86.24 \pm 1.24$ | $82.88 \pm 1.23$ | $87.20 \pm 0.40$ |
| ✓ | ✓ | ✓ | $\mathbf{86.45 \pm 1.15}$ | $\mathbf{88.26 \pm 0.97}$ | $\mathbf{85.26 \pm 0.36}$ | $\mathbf{88.52 \pm 0.20}$ |

### A.4.5 RESISTANCE OF PCFI TO MISSING FEATURES

To verify the resistance of PCFI against missing features, we compared the averaged node classification accuracy for the 6 datasets by increasing $r_m$ as shown in Table. 8. We compared each average accuracy with different $r_m$ from $r_m = 0$ to $r_m = 0.95$. For structural missing, PCFI loses only 2.23% of relative average accuracy with $r_m = 0.9$, and 4.82% with $r_m = 0.995$. In the case of uniform missing, PCFI loses only 1.81% of relative average accuracy with $r_m = 0.9$, and 2.66% with $r_m = 0.995$. Note that classification with $r_m = 0.995$ is the extremely missing case in which only 0.5% of features are known. This result demonstrates that PCFI is robust to missing features.

Even at the same $r_m$, we observed the average accuracy for structural missing is lower than one for uniform missing. We analyzed this observation in the aspect of confidence. Since features are missing in a node unit for structural missing, all channel features of a node have the same SPD-S, i.e., $\boldsymbol{S}_{i,1} = \ldots = \boldsymbol{S}_{i,F}$. Hence, in a node far from its nearest source node, every channel feature has low confidence. Then, every missing feature of the node can not be improved via node-wise inter-channel propagation due to the absence of highly confident channel features in the node.

Table 8: Node classification accuracy (%) of PCFI at different missing rates of for two missing types. For each experiment, we report the mean with standard deviation (mean $\pm$ std). For each missing type, we report average accuracy with relative drop (%p) compared to a full-feature setting (average **(drop)**).

| Missing type | Dataset | Full features | 50% missing | 90% missing | 99.5% missing |
|---|---|---|---|---|---|
| Structural missing | Cora | $82.35 \pm 1.49$ | $80.37 \pm 1.55$ | $78.88 \pm 1.43$ | $75.49 \pm 2.10$ |
| | CiteSeer | $70.98 \pm 1.46$ | $70.10 \pm 2.02$ | $69.76 \pm 1.96$ | $66.18 \pm 2.75$ |
| | PubMed | $77.49 \pm 2.05$ | $75.93 \pm 1.44$ | $76.12 \pm 1.87$ | $74.66 \pm 2.26$ |
| | Photo | $92.14 \pm 0.62$ | $91.81 \pm 0.54$ | $89.96 \pm 0.68$ | $87.70 \pm 1.29$ |
| | Computers | $85.67 \pm 1.41$ | $84.91 \pm 0.88$ | $82.40 \pm 1.38$ | $79.25 \pm 1.19$ |
| | OGBN-Arxiv | $72.28 \pm 0.11$ | $71.64 \pm 0.19$ | $70.39 \pm 0.19$ | $68.72 \pm 0.28$ |
| | Average | $80.15$ | $79.13\,(-\mathbf{1.02})$ | $77.92\,(-\mathbf{2.23})$ | $75.33\,(-\mathbf{4.82})$ |
| Uniform missing | Cora | $82.35 \pm 1.49$ | $81.28 \pm 1.59$ | $79.55 \pm 1.32$ | $78.53 \pm 1.39$ |
| | CiteSeer | $70.98 \pm 1.46$ | $71.68 \pm 1.92$ | $69.92 \pm 1.68$ | $69.40 \pm 1.85$ |
| | PubMed | $77.49 \pm 2.05$ | $76.88 \pm 2.09$ | $76.56 \pm 2.08$ | $76.44 \pm 1.64$ |
| | Photo | $92.14 \pm 0.62$ | $91.83 \pm 0.58$ | $89.84 \pm 1.00$ | $88.60 \pm 1.30$ |
| | Computers | $85.67 \pm 1.41$ | $84.96 \pm 1.15$ | $83.14 \pm 0.72$ | $81.79 \pm 0.70$ |
| | OGBN-Arxiv | $72.28 \pm 0.11$ | $71.78 \pm 0.09$ | $70.91 \pm 0.17$ | $70.19 \pm 0.15$ |
| | Average | $80.15$ | $79.73\,(-\mathbf{0.42})$ | $78.34\,(-\mathbf{1.81})$ | $77.49\,(-\mathbf{2.66})$ |

Thus, the node is likely to be misclassified. In contrast, for uniform missing, channel features of a node have various PC where known channel features have high confidence and propagate their feature information to unknown channel features in the node. Hence, most nodes can be classified well owing to the recovered features from highly confident channel features. We claim that this observation shows the validity of the concept of channel-wise confidence. The tables containing accuracy at different $r_m$ for the two missing types is in Section A.5.

### A.4.6 Classification Accuracy According to Pseudo-Confidence

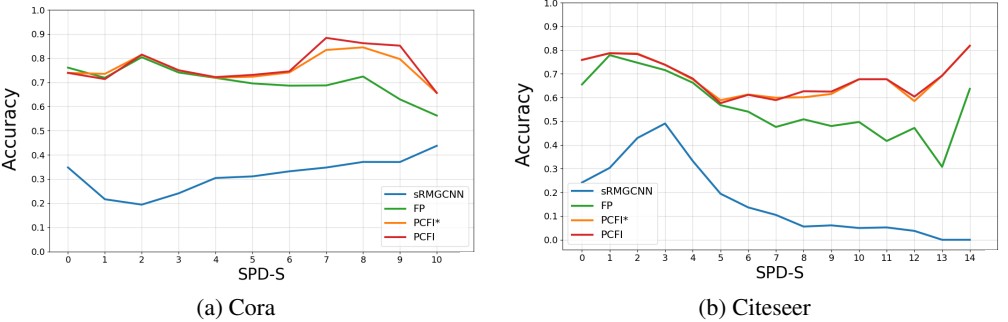

(a) Cora                    (b) Citeseer

Figure 6: Node classification accuracy (%) according to SPD-S of test nodes. For both Cora and CiteSeer, structural missing with $r_m = 0.995$ is applied. PCFI* denotes PCFI without node-wise inter-channel propagation. PCFI shows a noticeable performance gain especially for nodes with low-PC missing features (large-SPD-S nodes). Also, node-wise inter-channel propagation shows its effectiveness on nodes with low-PC features.

Under a structural missing setting, node features in a node have the same SPD-S, i.e., $\boldsymbol{S}_{i,1} = ... = \boldsymbol{S}_{i,d}$ for the $i$-th node. Since pseudo confidence $\xi_{i,d}$ is calculated by $\xi_{i,d} = \alpha^{\boldsymbol{S}_{i,d}}$, node features within a node also have the same pseudo-confidence (PC) for structural missing, i.e., $\xi_{i,1} = ... = \xi_{i,d}$. We refer to SPD-S of node features within a node as SPD-S of the node. Similarly, we refer to PC of node features with a node as PC of the node. The test nodes are divided according to SPD-S of the nodes. Then, to observe the relationship between PC and classification accuracy, we calculated classification accuracy of nodes in each group. We conducted experiments on Cora and CiteSeer under a structural missing setting with $r_m = 0.995$. We compared PCFI with sRMGCNN and FP.

Figure 6 shows node classification accuracy according to SPD-S of test nodes. For both datasets, as SPD-S of nodes increases, the accuracy of FP tends to decrease. However, for large-SPD-S nodes, PCFI gains noticeable performance improvement compared to FP. Furthermore, the results on Cora show that PCFI outperforms PCFI without node-wise inter-channel propagation on large-SPD-S nodes. Since large SPD-S means low PC, we can observe that PCFI imputes low-PC missing features effectively.

### A.4.7    DEGREE OF FEATURE RECOVERY ACCORDING TO PSEUDO-CONFIDENCE

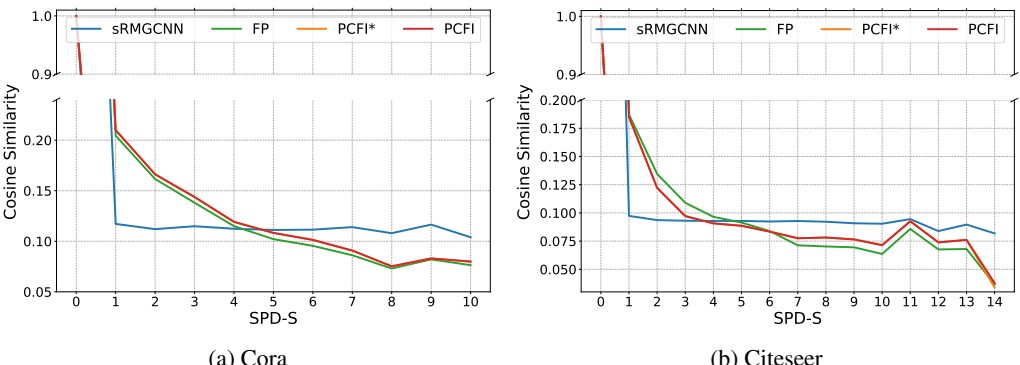

(a) Cora                                                        (b) Citeseer

Figure 7: Cosine similarity between $X_{i,:}$ (original features in a node) and $\hat{X}_{i,:}$ (its recovered features) according to SPD-S of features within the node. For both datasets, we randomly selected 99.5% nodes and remove all features of the selected nodes (structural missing) so that all channel features within a node have the same SPD-S. The imputed feature similarity between two nodes tends to decrease as the shortest path distance between the two nodes increases.

We further conducted experiments to observe the degree of feature recovery according to SPD-S of nodes. To evaluate the degree of feature recovery for each node, we measured the cosine similarity between recovered node features and original node features. The setting for experiments is the same as in section A.4.6.

Figure 7 demonstrates the results on Cora and CiteSeer. As SPD-S of nodes increases from zero, which means PC of nodes decreases, the cosine similarity between tends to decreases. In other words, PC of nodes increases, the cosine similarity tends to increase. This shows that the pseudo-confidence is designed properly based on SPD-S, which reflect the confidence.

Unlike the tendency in node classification accuracy, PCFI shows almost the same degree of feature recovery as PCFI without node-wise inter-channel propagation. This means that node-wise inter-channel propagation improves performance with very little refinement. That is, higher classification accuracy on nodes does not necessarily mean higher degree of feature recovery of the nodes. We leave a detailed analysis of this observation for future work.

### A.4.8 Computational Cost

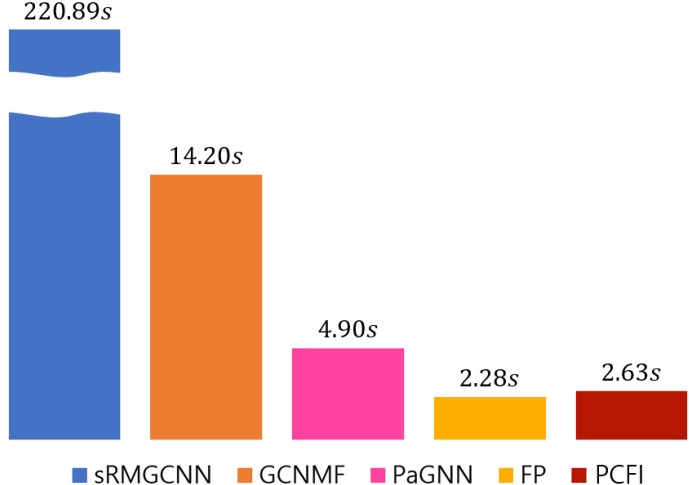

Figure 8: Training time (in seconds) of methods on Cora under a structural-missing setting with $r_m = 0.995$.

To compare computational cost, we measured the total training time on a single split of Cora. We performed node classification under a structural-missing setting with $r_m = 0.995$. The training time of each method is shown in Figure 8. The training time for the feature imputation methods (sRMGCNN, FP, PCFI) includes both the time for feature imputation and training of a downstream GCN. PCFI shows less training time than the other methods except for FP. Even compared to FP, PCFI takes only 15.4% more time than FP. For PCFI under a uniform-missing setting, the time for computing SPD-S increases by the number of channels. PCFI outperforms the state-of-the-art methods with less or comparable computational cost to the other methods.

## A.5 NODE CLASSIFICATION ACCURACY ACCORDING TO MISSING RATE FOR EACH SETTING

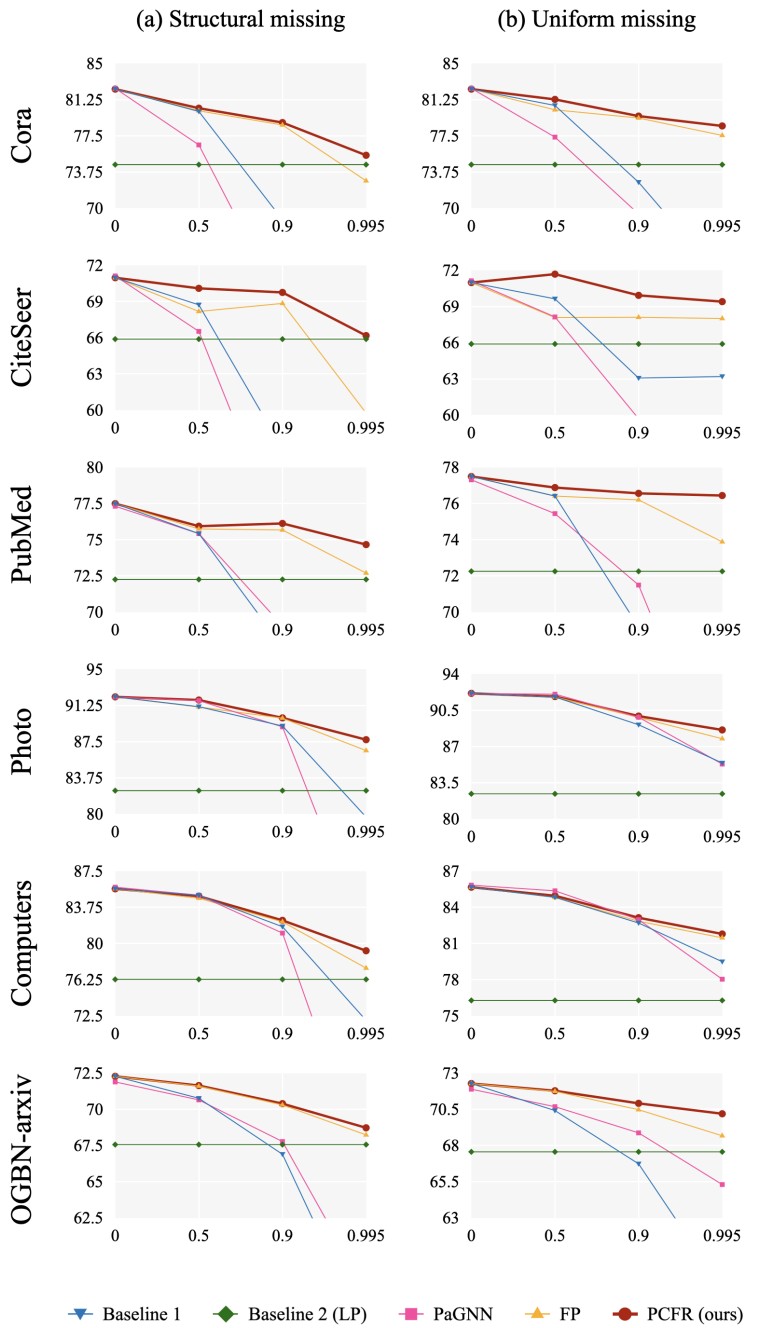

Figure 9: Average accuracy (%) for each setting (missing type, dataset).

A.6   PyTorch-style Pseudo-code of Pseudo-Confidence-based Feature Imputation (PCFI)

---

**Algorithm 1** PyTorch-style pseudo-code of PCFI

---

```
class PCFI(torch.nn.Module):
    def __init__(self, K, alpha, beta):
        super(PCFI, self).__init__()
        self.K = K
        self.alpha = alpha
        self.beta = beta

    # edge_index has shape [2, |E|]
    # mask is a boolean tensor indicating known features
    # with True
    def propagate(self, x, edge_index, mask, mask_type):
        nv, feat_dim = x.shape
        # Channel-wise inter-node diffusion
        out = torch.zeros_like(x)
        out[mask] = x[mask]
        # structural missing case
        if mask_type == "structural":
            SPDS = self.compute_SPDS(edge_index, mask, mask_type)
            Wbar = self.compute_Wbar(edge_index, mask, mask_type)
            for t in range(self.K):
                out = torch.sparse.mm(Wbar, out)
                out[mask] = x[mask]
            SPDS = SPDS.repeat(feat_dim, 1)
        # uniform missing case
        if mask_type == "uniform":
            SPDS = self.compute_SPDS(edge_index, mask, \
                                        mask_type, feat_dim)
            for d in range(feat_dim):
                Wbar = self.compute_Wbar(edge_index, SPDS[d])
                for i in range(self.K):
                    out[:,d] = torch.sparse.mm(Wbar, \
                             out[:,d].reshape(-1,1)).reshape(-1)
                    out[mask[:,d],d] = x[mask[:,d],d]
        # Node-wise inter-channel propagation
        cor = torch.corrcoef(out.T).nan_to_num().fill_diagonal_(0)
        a1 = (self.alpha ** SPDS.T) * (out - torch.mean(out,\
                                                   dim=0))
        a2 = torch.matmul(a1, cor)
        out1 = self.beta * (1 - self.alpha ** SPDS.T) * a2
        out = out + out1
        return out
```

---

**Algorithm 2** PyTorch-style pseudo-code for SPD-S

```python
# get SPD-S of each node by computing the k-hop subgraph
# around the source nodes
# In this code, SPDS has shape [F, N]
def compute_SPDS(self, edge_index, feature_mask, mask_type, \
                                            feat_dim = None):
    nv = feature_mask.shape[0]
    # structural missing case
    if mask_type == "structural":
        SPDS = torch.zeros(nv, dtype = torch.int)
        # v_0 represents the source nodes
        v_0 = torch.nonzero(feature_mask[:, 0]).view(-1)
        v0_to_k = v_0
        SPDS[v_0] = 0
        k = 1
        while True:
            v_k_hop_sub = torch.geometric.utils \
                .k_hop_subgraph(v_0, k, edge_index, \
                num_nodes = nv)[0]
            # v_k represents the nodes of which SPS-S = k
            v_k = torch.from_numpy(np.setdiff1d(v_k_hop_sub, \
                                        v_0_to_k))
            if len(v_k) == 0:
                break
            SPDS[v_k] = k
            torch.cat([v_0_to_k, v_k], dim=0)
            k += 1
    # uniform missing case
    if mask_type == "uniform":
        SPDS = torch.zeros(feat_dim, nv)
        for d in range(feat_dim):
            v_0 = torch.nonzero(feature_mask[:,d]).view(-1)
            v_0_to_k = v_0
            SPDS[d, v_0] = 0
            k = 1
            while True:
                v_k_hop_sub = \
                torch.geometric.utils.k_hop_subgraph( \
                        v_0, k, edge_index, num_nodes = nv)[0]
                v_k = torch.from_numpy(np.setdiff1d \
                            (v_k_hop_sub, v_0_to_k))
                if len(v_k) == 0:
                    break
                SPDS[d, v_k] = k
                torch.cat([v_0_to_k, v_k], dim=0)
                k += 1
    return SPDS
```

—

---

**Algorithm 3** PyTorch-style pseudo-code for $\overline{W}^{(d)}$

---

```
def compute_Wbar(self, edge_index, SPDS):
    # In the propagation, row and column correspond to
    # destination and source, respectively.
    row, col = edge_index[0], edge_index[1]
    SPDS_row = SPDS[row]
    SPDS_col = SPDS[col]
    edge_weight = self.alpha ** (SPDS_col - SPDS_row + 1)
    # row-wise sum
    deg_W = torch_scatter.scatter_add(edge_weight, row, \
                                    dim_size= SPDS.shape[0])
    # normalize W^(d) to row-stochastic Wbar^(d)
    deg_W_inv = deg_W.pow_(-1.0)
    deg_W_inv.masked_fill_(deg_W_inv == float("inf"), 0)
    norm_edge_weight = edge_weight * deg_W_inv[row]
    Wbar = torch.sparse.FloatTensor(edge_index, \
                                    values= norm_edge_weight)
    return Wbar
```

---

