# OpenReview forum: "Confidence-Based Feature Imputation for Graphs with Partially Known Features"
_ICLR.cc/2023/Conference — ICLR 2023 poster_

### Official Review · Reviewer_tfst · 2022-10-20

**Confidence:** 1
**Correctness:** 3
**Technical Novelty And Significance:** 2
**Empirical Novelty And Significance:** 2
**Recommendation:** 6

**Clarity, Quality, Novelty And Reproducibility:**

Overall, the paper is clearly written.
The quality of the methods looks sound, and the novelty is fair.

**Strength And Weaknesses:**

Strength :

The problem is clearly described and the methods are well presented.
The scheme can even handle at an exceedingly high missing rate (e.g., 99.5%) and achieves state-of-the-art accuracy for both semi-supervised node classification and link prediction on various datasets containing a high rate of missing features.

Weaknesses:
I did not find notable weaknesses of this paper.

**Summary Of The Paper:**

The paper targets problems with graph learning tasks containing missing node features.
Their key idea is to assign different pseudo-confidence to each imputed channel features.
The proposed imputation scheme includes two processes.
The first one is the feature imputation using channel-wise inter-node diffusion, and the second is the feature refinement via node-wise inter-channel propagation.

**Summary Of The Review:**

The work motivates by an interesting and practical problem when the graph learning is conducted in presence of missing node features.
Overall, the methods look sound and reasonable.

---

> ### Author Response · Authors · 2022-11-16
> **Response to Reviewer tfst**
>
> We appreciate your encouragement and hope that the following supplementary explanation can make you more confident.
>
> Gathering complete data is very expensive, and only a few users disclose their information entirely on social media. Thus there have been many efforts to deal with graphs with missing features [1, 2, 3, 4]. Prior works recover missing features by aggregating features from connected nodes with equal importance. However, we assign different importance to each missing feature. Since linked nodes tend to have similar features in many graphs [5], we assume that the quality of a recovered feature is different according to the distance from observed features. From this motivation, we define pseudo-confidence, which encodes the distance between a set of observed features and a missing feature. Pseudo-confidence plays a crucial role in two stages (channel-wise inter-node diffusion and node-wise inter-channel propagation) that form our method. To the best of our knowledge, our method is the first attempt that adopts the confidence of a missing feature for graph imputation.
>
> Unlike FP [1], channel-wise inter-node diffusion introduces bidirected edges that can strengthen a message passing from a high-confidence feature to a low-confidence feature and weaken a message passing in the reverse direction. Then, node-wise inter-channel propagation further refines the recovered features with consideration of inter-channel dependency which is overlooked by FP. Pseudo-confidence and inter-channel dependency enable our method to squeeze out the remaining information in graphs with missing features. We demonstrate our method outperforms the state-of-the-art methods on two main graph learning tasks which are node classification and link prediction (in Table 1 and Table 2 of the manuscript) and have conducted further experiments in this rebuttal period to analyze the performance of pseudo-confidence-based feature imputation (PCFI) on graphs with different feature homophily (see the response to Reviewer NNts).  This result shows that our method has a significant advantage over the state-of-the-art method on high-homophily graphs.
>
> [1] Rossi, Emanuele, et al. "On the unreasonable effectiveness of feature propagation in learning on graphs with missing node features." arXiv preprint arXiv:2111.12128 (2021).
>
> [2] Taguchi, Hibiki, Xin Liu, and Tsuyoshi Murata. "Graph convolutional networks for graphs containing missing features." Future Generation Computer Systems 117 (2021): 155-168.
>
> [3] Chen, Xu, et al. "Learning on attribute-missing graphs." IEEE transactions on pattern analysis and machine intelligence (2020).
>
> [4] Jiang, Bo, and Ziyan Zhang. "Incomplete graph representation and learning via partial graph neural networks." arXiv preprint arXiv:2003.10130 (2020).
>
> [5] McPherson, Miller, Lynn Smith-Lovin, and James M. Cook. "Birds of a feather: Homophily in social networks." Annual review of sociology (2001): 415-444.

---

### Official Review · Reviewer_NNts · 2022-10-25

**Confidence:** 3
**Correctness:** 4
**Technical Novelty And Significance:** 3
**Empirical Novelty And Significance:** 4
**Recommendation:** 8

**Clarity, Quality, Novelty And Reproducibility:**

The paper is well-written, and provides enough detail to reproduce the experiments.

**Strength And Weaknesses:**

Strength: The idea is simple and easy to understand. Performance on the 0.5% sampling setting is impressive. Notations are explicitly defined and derivations are explained well. There are enough implementation details to reproduce the experiments. Extensive hyperparameter search was done (reported in appendix).

Weakness: The advantage compared to feature propagation (FP) is not yet convincing.
Since some of the concepts are straightforward, some sections (e.g. section 3.4) could be simplified to make room for more experiments that go beyond the benchmark datasets. For example, an experiment with simulated data to understand the model's behavior better could have made the paper's argument stronger.

**Summary Of The Paper:**

This paper proposes an imputation method for missing node features in graph neural networks. There is a node-wise diffusion as well as channel-wise diffusion that is based on the "pseudo confidence" of a missing feature node, which is inversely proportional to the distance to the closest observed feature node. Semi-supervied learning and link prediction experiments are conducted with good results in the very low sampling regime (0.5%).

**Summary Of The Review:**

I think the idea presented in this paper is interesting, and it is explained well. There are also extensive experiments with promising results. But it's not clear why this ~1% gain in accuracy makes this better than the very simple FP.

---

> ### Author Response · Authors · 2022-11-16
> **Response to Reviewer NNts [Part 2/2]**
>
> $\textbf{Q2.}$ I think the idea presented in this paper is interesting, and it is explained well. There are also extensive experiments with promising results. But it's not clear why this ~1% gain in accuracy makes this better than the very simple FP.
>
> $\textbf{A2.}$ Graph learning with missing features aims to prevent performance drop compared to fully observed settings. We conducted experiments with only 0.5% of features by exploiting little remaining information. The table below shows the average performance drop across all the used datasets for each experiment at missing rate $r_m = 0.995$. In terms of absolute values, there are ~2%p gains of ours over FP. Moreover, compared to the very simple FP, our method increases training time slightly (in Sec. A.4.8 in the revised manuscript).
>
> | task         |        node       |   classification  |                  |       link       |    prediction    |                  |
> |--------------|:-----------------:|:-----------------:|:----------------:|:----------------:|:----------------:|:----------------:|
> | missing type |     structural    |      uniform      |    structural    |    structural    |      uniform     |      uniform     |
> |              | $\Delta$ Acc (\%p) | $\Delta$ Acc (\%p) | $\Delta$ AP (\%p) | $\Delta$ AUC (\%p) | $\Delta$ AP (\%p) | $\Delta$ AUC (\%p) |
> | FP           |      $-7.23$      |      $-3.93$      |      $-8.17$     |      $-6.23$     |      $-6.52$     |      $-4.41$     |
> | PCFI         |      $-4.82$      |      $-2.66$      |      $-5.53$     |      $-3.89$     |      $-3.86$     |      $-2.47$     |
> | gain         |       $2.41$      |       $1.27$      |      $2.64$      |      $2.34$      |      $2.66$      |      $1.94$      |

---

> > ### Comment · Reviewer_NNts · 2022-12-02
> > **acknowledgement of comments**
> >
> > I apologize for the delay. Thank you for the detailed experimental results that show the proposed method's advantage over FP. I have increased my score.

---

> > > ### Author Response · Authors · 2022-12-03
> > > **Second Response to Reviewer NNts**
> > >
> > > We thank your decision. Your advice has helped us demonstrate our method's advantage over FP more clearly.

---

> ### Author Response · Authors · 2022-11-16
> **Response to Reviewer NNts [Part 1/2]**
>
> $\textbf{Q1.}$ Weakness: The advantage compared to feature propagation (FP) is not yet convincing. Since some of the concepts are straightforward, some sections (e.g. section 3.4) could be simplified to make room for more experiments that go beyond the benchmark datasets. For example, an experiment with simulated data to understand the model's behavior better could have made the paper's argument stronger.
>
> $\textbf{A1.}$ We agree with you that an experiment with simulated data to understand the model’s behavior better could make our argument stronger. Since pseudo-confidence is based on the assumption that features of linked nodes tend to be similar (i.e., assumes feature homophily), we conducted further experiments to analyze the performance of pseudo-confidence-based feature imputation (PCFI) on graphs with different feature homophily. Moreover, we explored under what circumstances the gain of PCFI over FP is maximized by a large margin with simulated data. The detailed experimental setup is specified at the end of the A1.
>
> The tables below show the node classification accuracy of experiments on graphs with different feature homophily (We report the same results in Figure 5 in Sec. A.4.3 of the revised manuscript.). Note that the Dirichlet energy ($E_D$) is used as a criterion by which homophily can be compared (increasing $-\log(E_D)$ means increasing feature homophily). As shown in the tables, the greater gain of PCFI over FP is obtained as feature homophily increases. Further, the proposed PCFI shows superior performance consistently regardless of feature homophily and the gain exceeds 10% on graphs with high feature homophily. Since our method is based on the assumption that linked nodes tend to have similar features, our method has a significant advantage over the state-of-the-art method on high-homophily graphs. Homophily is often inherent in many real-world networks [1]. Therefore, as the effectiveness of our method on various datasets is demonstrated in Table 1 and Table 2 of the manuscript, PCFI can be applied in many real-world situations.
>
> <Table A: Accuracy for graphs with class homophily $= 0.3$>
>
> | $-\log (E_D)$ |     PCFI (%)     |      FP (%)      | gain (%p) |
> |:-------------:|:----------------:|:----------------:|:--------:|
> |    $-7.32$    | $34.91 \pm 2.62$ | $31.02 \pm 2.80$ |  $3.89$  |
> |    $-6.71$    | $35.69 \pm 2.39$ | $32.58 \pm 1.99$ |  $3.11$  |
> |    $-6.12$    | $36.73 \pm 2.07$ | $30.83 \pm 2.41$ |  $5.90$  |
> |    $-5.51$    | $38.39 \pm 3.66$ | $32.93 \pm 4.49$ |  $5.46$  |
> |    $-4.93$    | $36.21 \pm 2.84$ | $29.45 \pm 3.60$ |  $6.76$  |
> |    $-4.38$    | $36.54 \pm 3.31$ | $28.11 \pm 3.73$ |  $8.43$  |
> |    $-3.95$    | $37.10 \pm 1.95$ | $26.78 \pm 2.72$ |  $10.32$ |
>
> <Table B: Accuracy for graphs with class homophily $= 0.5$>
>
> | $-\log (E_D)$ |     PCFI (%)     |      FP (%)      | gain (%p) |
> |:-------------:|:----------------:|:----------------:|:--------:|
> |    $-7.32$    | $64.19 \pm 2.48$ | $59.52 \pm 3.42$ |  $4.67$  |
> |    $-6.72$    | $65.73 \pm 3.21$ | $60.52 \pm 3.90$ |  $5.21$  |
> |    $-6.11$    | $66.09 \pm 3.22$ | $60.71 \pm 3.19$ |  $5.38$  |
> |    $-5.52$    | $66.61 \pm 2.62$ | $59.26 \pm 3.40$ |  $7.35$  |
> |    $-4.93$    | $65.76 \pm 2.76$ | $56.78 \pm 3.95$ |  $8.98$  |
> |    $-4.37$    | $67.77 \pm 2.47$ | $57.03 \pm 4.49$ |  $10.74$ |
> |    $-3.90$    | $66.87 \pm 1.99$ | $54.94 \pm 3.57$ |  $11.93$ |
>
> <Experimental setup>
>
> For graphs with different feature homophily, we first prepared two graphs (graphs with class homophily of 0.3 and 0.5) from the synthetic dataset which contains graphs with various class homophily [2]. The class homophily of the graphs indicates the likelihood of a node forming an edge to a node with the same class. While preserving the graph structure and class distribution, we newly assigned generated node features so that each generated graph has a different feature homophily. Then, we applied structural missing with missing rate $r_m = 0.995$. We randomly generated 10 different training/validation/test splits for node classification and we report the mean and standard deviation of classification accuracy. For a fair comparison, hyperparameters of both methods are searched in the manner specified in Sec. A.3.1 of the revised manuscript. Further details of the experimental setup including feature generation are presented in Sec. A.4.3 of the revised manuscript.
>
> Based on your constructive feedback, we added the experiment with the synthetic data in Sec. A.4.3 in the revised manuscript.
>
> [1] McPherson, Miller, Lynn Smith-Lovin, and James M. Cook. "Birds of a feather: Homophily in social networks." Annual review of sociology (2001): 415-444.
>
> [2] Abu-El-Haija, Sami, et al. "Mixhop: Higher-order graph convolutional architectures via sparsified neighborhood mixing." international conference on machine learning. PMLR, 2019.

---

### Official Review · Reviewer_z9Q8 · 2022-10-25

**Confidence:** 2
**Clarity, Quality, Novelty And Reproducibility:** See above.
**Correctness:** 4
**Technical Novelty And Significance:** 3
**Empirical Novelty And Significance:** 4
**Recommendation:** 8

**Strength And Weaknesses:**

Overall, the paper is understandable, the method is reasonable and the experimental results are good. As described below, I am not sure about the methodological novelty. The application impact seems high, although this is also hard to tell because the benchmarks used are on relatively contrived tasks.

My main concern is that I could not tell what parts of the proposed method are novel. This is partly because I am not too familiar with graph imputation. The proposed method contains many components and all components are described as though they are novel. In particular, the authors indicate that their main contribution as "pseudoconfidence" as alpha^S, where S is the distance in the graph.  They should explain that this is a common strategy used in random walk methods, Monte Carlo reinforcement learning, and many other methods. Similarly, they should explain and cite the inspiration for each other part of their method. It seems likely that there are novel components and novel ways in which preexisting components are combined, but I could not work out what these are.

What real-world problems have 99% missing features?

**Summary Of The Paper:**

The authors tackle the problem of imputation in graphs, in which we want to predict missing node features in graph-structured data. Their approach uses a strategy involving pseudo-confidence (PC), where the PC of a given feature is proportional to alpha^S, where S is the corresponding node's nearest labeled neighbor. They combine this with a propagation strategy which shares information between features of the same node and between neighboring nodes.

The authors evaluate their method on a benchmark data set OGN. They follow (Rossi et al, 2021) in their evaluation. Their primary evaluation is on a data set from arXiv, in which nodes represent papers, edges denote citations between papers and node features are a representation of the words used in the paper's title and abstract. They hold out a subset of features of each node and impute these held-out features. They show that on these tasks their method outperforms existing methods including GCNMF and Feature Propagation.

**Summary Of The Review:**

See above.

---

> ### Author Response · Authors · 2022-11-16
> **Response to Reviewer z9Q8**
>
> $\textbf{Q1-1.}$ My main concern is that I could not tell what parts of the proposed method are novel. This is partly because I am not too familiar with graph imputation. The proposed method contains many components and all components are described as though they are novel.
>
> $\textbf{Q1-2.}$ In particular, the authors indicate that their main contribution is "pseudo-confidence" as alpha^S, where S is the distance in the graph. They should explain that this is a common strategy used in random walk methods, Monte Carlo reinforcement learning, and many other methods.
>
> $\textbf{Q1-3.}$ Similarly, they should explain and cite the inspiration for each other part of their method. It seems likely that there are novel components and novel ways in which preexisting components are combined, but I could not work out what these are.
>
> $\textbf{A1.}$ Our novel components are pseudo-confidence, channel-wise inter-node diffusion, and node-wise inter-channel propagation.
>
> Pseudo-confidence is a new concept to overcome the performance degradation of FP [1] taking graph diffusion through undirected edges. Our approach adopts bidirected edges that can strengthen a message passing from a high-confidence feature to a low-confidence feature and weaken a message passing in the reverse direction. To replace the unmeasurable confidence of a feature, we propose pseudo-confidence that encodes the distance between a set of observed features and a missing feature. As the reviewer mentioned, while using a distance can be a common strategy in other tasks, to the best of our knowledge, ours is the first model that leverages a distance for graph imputation, which brings significant performance enhancement. Based on this concept, we design two stages for imputation: channel-wise inter-node diffusion and node-wise inter-channel propagation.
>
> In the channel-wise inter-node diffusion, the strength of channel-wise message passing (edge weight) is assigned differently depending on the passing direction, unlike FP. To this end, we design an assigning rule of bidirected edge weights based on pseudo-confidence, as presented in Proposition 1.  Hence, in our method, the strength of a message passing from a high-pseudo-confidence feature to a low-pseudo-confidence feature becomes stronger than that in the reverse direction. This bidirectional message passing with different strengths is in contrast to FP where a message passing between two nodes is done with the same strength regardless of the direction.
>
> The node-wise inter-channel propagation is a totally new component considering inter-channel dependency via propagation within a node. In the case of high rates of missing features, a missing feature with low confidence can not be recovered via the channel-wise inter-node diffusion. To overcome this issue, we further refine this missing feature by using information propagated from highly correlated channel features with high confidence. The effectiveness of the proposed components has been verified through experiments on two main graph learning tasks (Table 1 and Table 2 in the manuscript).
>
> We uploaded the revised manuscript that reflects your constructive feedback.
>
> [1] Rossi, Emanuele, et al. "On the unreasonable effectiveness of feature propagation in learning on graphs with missing node features." arXiv preprint arXiv:2111.12128 (2021).
>
> $\textbf{Q2.}$ What real-world problems have 99% missing features?
>
> $\textbf{A2.}$ Social networks are practical examples of high rates of missing features. According to a survey [2], only 11% of young respondents do not activate any privacy setting. 300 million accounts on Instagram are estimated as private accounts [3]. Moreover, sensitive information such as age and income tends to have very low disclosure rates [4, 5]. For example, only 4.7% of respondents provide complete information related to personal economic status [6].  Since social media privacy concerns have spiked in recent years, the ratio of public information is expected to decrease. Such kinds of surveys conducted on social networks can be examples of more extremely missing features. A small number of subjects for surveys are sampled and a small rate of sampled subjects respond. Hence the surveys might contain extremely high rates of missing features.
>
>
> [2] https://www.viasatsavings.com/news/blog/are-more-people-public-or-private-on-social-media/#twitter
>
> [3] Gaffney, D. "Estimating Instagram’s Actual Population Statistics." (2016).
>
> [4] Nulty, Duncan D. "The adequacy of response rates to online and paper surveys: what can be done?." Assessment & evaluation in higher education 33.3 (2008): 301-314.
>
> [5] Bollinger, Christopher R., and Barry T. Hirsch. "Is earnings nonresponse ignorable?." Review of Economics and Statistics 95.2 (2013): 407-416.
>
> [6] Lan, Wei, et al. "Imputations for high missing rate data in covariates via semi-supervised learning approach." Journal of Business & Economic Statistics 40.3 (2022): 1282-1290.

---

> > ### Comment · Reviewer_z9Q8 · 2022-11-20
> > **2022-11-20**
> >
> > Thank you for the response. These comments substantially address my concerns. I change my score to 8. My confidence remains low, as this topic is a bit out of my area of expertise.

---

> > > ### Author Response · Authors · 2022-11-21
> > > **Second Response to Reviewer z9Q8**
> > >
> > > We thank your decision. Your insightful feedback has further improved our paper.

---

> > > ### Author Response · Authors · 2022-11-23
> > > **A gentle reminder for Reviewer z9Q8**
> > >
> > > Dear Reviewer z9Q8,
> > >
> > > We found out that your score has not changed yet on the system. We respectfully ask if you could update your score on the system.
> > >
> > > Thank you.

---

> > > > ### Comment · Reviewer_z9Q8 · 2022-11-23
> > > > **Fixed**
> > > >
> > > > Fixed

---

### Author Response · Authors · 2022-11-17
**General Response to Reviewers**

Dear Reviewers,

We appreciate your solid feedback for improving our paper. Since we have addressed each review separately, we summarize the main changes as follows:

1.  We added node classification results on simulated data, which clearly demonstrate an advantage over FP (in Sec. A.4.3 in the revised manuscript). For graphs with high feature homophily, our method shows more than 10%p gain compared to FP.
2.  We revised Sec. 1 and Sec. 3.4 to show that there is a clear novelty compared to existing methods.

We hope that the reviewers find our response convincing, and please let us know if the raised issues are addressed. If the reviewers have more concerns, we would like to have an opportunity to further address them. We uploaded the revised manuscript. The modified parts are marked in blue.

Thank you.

---

### Decision · Program_Chairs · 2023-01-20

**Decision:**

Accept: poster

**Justification For Why Not Higher Score:**

While the paper makes some exciting contributions, these might not be ground breaking enough for a spotlight. However, I am open to raising my score if recalibration is needed.

**Justification For Why Not Lower Score:**

The paper makes novel contributions on an important topic, the presentation is clear and the authors have satisfactorily address all the concerns raised by the reviewers.

**Metareview: Summary, Strengths And Weaknesses:**

The paper considers the problem of node feature imputation in graphs and introduces a confidence measure that is assigned to each imputed feature. The robustness of the approach is demonstrated on semi-supervised node classification and link prediction, where it is shown to endure high rate of missing features.

The reviewers and AC all agree that the approach presented in this paper is intuitive, interesting and promising, and that the exposition is very clear.  The empirical evaluation is extensive and shed light on many interesting aspects of the approach. The author feedback clarifying novelty is very much appreciated. In addition, the new experiment convincingly demonstrate the superiority of the proposed approach.

As future work, it would be interesting to study the "active learning" setting where the confidence measure could be used to prioritize which missing features to acquire. See e.g.  Desjardins, Marie, James MacGlashan, and Kiri L. Wagstaff. "Confidence-based feature acquisition to minimize training and test costs." Proceedings of the 2010 SIAM International Conference on Data Mining. Society for Industrial and Applied Mathematics, 2010.

**Note From Pc:**

if the above contains the word "oral" or "spotlight" please see: "oral" presentation means -> notable-top-5% and "spotlight" means -> notable-top-25%. As stated in our emails, we are disassociating presentation type from AC recommendations